# A requirement of Polo-like kinase 1 in murine embryonic myogenesis and adult muscle regeneration

Zhihao Jia[1†], Yaohui Nie[1,2†], Feng Yue[1], Yifan Kong[1], Lijie Gu[1], Timothy P Gavin[2], Xiaoqi Liu[3,4], Shihuan Kuang[1,4]*

[1]Department of Animal Sciences, Purdue University, West Lafayette, United States; [2]Department of Health and Kinesiology, Purdue University, West Lafayette, United States; [3]Department of Biochemistry, Purdue University, West Lafayette, United States; [4]Center for Cancer Research, Purdue University, West Lafayette, United States

**Abstract** Muscle development and regeneration require delicate cell cycle regulation of embryonic myoblasts and adult muscle satellite cells (MuSCs). Through analysis of the Polo-like kinase (Plk) family cell-cycle regulators in mice, we show that Plk1's expression closely mirrors myoblast dynamics during embryonic and postnatal myogenesis. Cell-specific deletion of *Plk1* in embryonic myoblasts leads to depletion of myoblasts, developmental failure and prenatal lethality. Postnatal deletion of *Plk1* in MuSCs does not perturb their quiescence but depletes activated MuSCs as they enter the cell cycle, leading to regenerative failure. The *Plk1*-null MuSCs are arrested at the M-phase, accumulate DNA damage, and apoptose. Mechanistically, *Plk1* deletion upregulates p53, and inhibition of p53 promotes survival of the *Plk1*-null myoblasts. Pharmacological inhibition of Plk1 similarly inhibits proliferation but promotes differentiation of myoblasts in vitro, and blocks muscle regeneration in vivo. These results reveal for the first time an indispensable role of Plk1 in developmental and regenerative myogenesis.
DOI: https://doi.org/10.7554/eLife.47097.001

*For correspondence:
skuang@purdue.edu

†These authors contributed
equally to this work

Competing interests: The
authors declare that no
competing interests exist.

Reviewing editor: Lynn
Megeney,

## Introduction

Adult skeletal muscle has an efficient regenerative capacity in response to muscle injury or physiological stimuli (i.e. intense exercise training). Muscle regeneration relies on a population of muscle resident stem cells, known as muscle satellite cells (MuSCs). These cells are located in a unique niche between the basal lamina and the plasma membrane of myofibers (*Mauro, 1961*) and remain in a quiescent state until activated by regenerative signals. Upon activation, MuSCs proliferate to generate a pool of myoblasts that eventually return to the quiescent state to replenish the MuSCs pool or differentiate to repair muscle injuries. Thus, cell cycle regulation of MuSCs is critical for precise control of the number of myoblasts that is needed for muscle regeneration. Reduced regenerative capacity was observed when a muscle was exposed to an inhibitor of mitotic division, colchicine (*Pietsch, 1961*) or irradiation (*Quinlan et al., 1995*). Knockout of *Cdkn1b*, *Cdkn1c* or *Rb* (negative cell cycle regulators) results in aberrant satellite cell activation and proliferation (*Chakkalakal et al., 2014*; *Hosoyama et al., 2011*; *Mademtzoglou et al., 2018*). Positive cell cycle regulators, including cyclin A, B, D, E, F and G, are upregulated in activated MuSCs to regulate cell cycle progression (*Cenciarelli et al., 1999*; *De Luca et al., 2013*; *Fukada et al., 2007*). Deregulation of cell cycle regulators p16 and p21, and Notch signaling in quiescent MuSCs in old mice leads to proliferative senescence, accumulation of DNA damage, mitotic catastrophe and high frequency of cell death (*Li et al., 2015*; *Liu et al., 2018*).

**eLife digest** Muscles have their own population of stem cells, called muscle satellite cells. These cells are essential for muscle growth and repair. In healthy adult muscles, they spend most of their time inactive, but when there is an injury, they reawaken and start dividing. Some of the new cells return to an inactive stem cell state to await the next injury. The rest mature into new muscle cells or join with damaged muscle fibres to help them repair.

The cell cycle is the series of events that a cell goes through from its birth until it divides. In muscle satellite cells, progression through the cell cycle is tightly controlled to ensure they divide and grow the correct amount. One of the proteins responsible for controlling the cell cycle is Polo-Like Kinase 1 (PLK1), but studying this protein is difficult. A common way to investigate a protein's effect is to delete the gene that makes it and observe the consequences. However, PLK1 is so essential to life that yeast, flies, zebrafish and mice all die when the gene is missing. Jia et al. deleted the gene that makes PLK1 only in mouse muscle satellite cells to find out the role this protein plays in controlling the cell cycle in stem cells.

Deleting the gene that codes for PLK1 before the mice were born was lethal. The embryos failed to develop mature muscle fibres, and they died. But deleting the gene after the mice were born had a different effect. The muscles developed normally, but they were unable to heal when injured. The same healing problem also happened when healthy mice received a drug that blocked the function of PLK1 protein. A closer look at the muscle satellite cells revealed the source of the problem. Without PLK1, the cells got stuck part way through their cell cycle, just before they were due to divide. They tried to become muscle cells, but they did not make it. Instead, the muscle satellite cells started to act as though their DNA had been damaged, and then they self-destructed.

Muscle satellite cells become less able to divide as we get older. They can also malfunction in some types of degenerative muscle diseases. Understanding how muscle satellite cells control their cell cycle could help us to find out what causes them to go wrong. Further work to understand PLK1 also has potential implications for cancer treatment. PLK1 blockers have been used to stop cancer cells from dividing, but Jia et al.'s findings show that this kind of drug may also hamper the ability of muscle to repair damage.

DOI: https://doi.org/10.7554/eLife.47097.002

The polo-like kinases (PLKs) are a conserved subfamily of Ser/Thr protein kinases that play pivotal roles in cell cycle regulation. The PLK family contains five members (PLK1-5) in mammals, all except for PLK5 contain an amino-terminal Ser/Thr kinase domain (*Archambault and Glover, 2009*; *de Cárcer et al., 2011*; *Liu, 2015*). Among the PLK kinases, PLK1 is the most conserved and best known for its role in mitosis via phosphorylation of different substrates (*Barr et al., 2004*). PLK1 also participates in modulating DNA replication and DNA damage checkpoints (*Takaki et al., 2008*). Overexpression of PLK1 is observed in several human tumors, including prostate and ovarian cancers, and muscle cell-derived rhabdomyosarcoma (*Hugle et al., 2015*; *van Vugt and Medema, 2005*). Inhibition of PLK1 by small interfering RNA or pharmacological inhibitors exerts antitumor effect in vitro and in vivo, providing strong preclinical and clinical support for the use of PLK1 inhibitors in cancer therapy (*Degenhardt and Lampkin, 2010*; *Lens et al., 2010*). Outside the cancer field, the role of PLK1 in normal mitotic cells especially stem cells are poorly understood. As *Plk1* gene deletion leads to embryonic lethality in mice, zebrafish, *Drosophila* and yeast (*Jeong et al., 2010*; *Lu et al., 2008*; *Ohkura et al., 1995*; *Sunkel and Glover, 1988*), conditional deletion of *Plk1* is necessary to understand its tissue or cell-type-specific functions. In this study, we used myogenic cell-specific targeted mutation to show that Plk1 is absolutely required for mitosis and survival of myogenic cells during muscle development and regeneration in mice.

## Results

### *Plk1* is dynamically expressed during muscle regeneration and myogenesis

To establish the relevance of Polo-like kinases in myogenesis, we surveyed the expression of Plk1–Plk4 (Plk5 was not surveyed as it does not have a kinase domain) at various time points during CTX-induced muscle regeneration. Activation and proliferation of MuSCs peaks at 3 days post injury (DPI), and the overall architecture of the muscle is restored by 10 DPI (*CHARGÉ and Rudnicki, 2004*). The mRNA levels of *Plk1*, *Plk3*, and *Plk4* were all transiently up-regulated after muscle injury, reaching peak expression levels at 3 DPI and returning to the preinjury levels at 10 DPI, but *Plk1* exhibited the most prominent fold change (increased by >13 fold) at 3 DPI (*Figure 1A*). The expression pattern of Plk1 corresponded to those of myogenic transcriptional factors Pax7 and MyoG at both mRNA (*Figure 1B*) and protein (*Figure 1C*) levels. We also surveyed the mRNA levels of *Plk1-4* during differentiation of primary myoblasts isolated from limb muscles. Compared with day 0 (proliferating myoblast), *Plk1*, *Plk3*, and *Plk4* levels were all down-regulated during myogenic differentiation (*Figure 1D*). Among these, *Plk1* and *Plk4* exhibited the most robust down-regulation (*Figure 1D*). The expression pattern of *Plk1* was inversely correlated to the expression of myogenic differentiation makers *Myog* and *eMyhc*, which were robustly upregulated during differentiation (*Figure 1E*). Consistently, *Plk1* levels progressively declined from embryonic day 17.5 (E17.5) to postnatal day 90 (P90) during limb muscle differentiation and maturation in vivo (*Figure 1F*). Since Plk1 is the most dynamically regulated Plks during myogenesis, we focused on Plk1 for the rest of the current study.

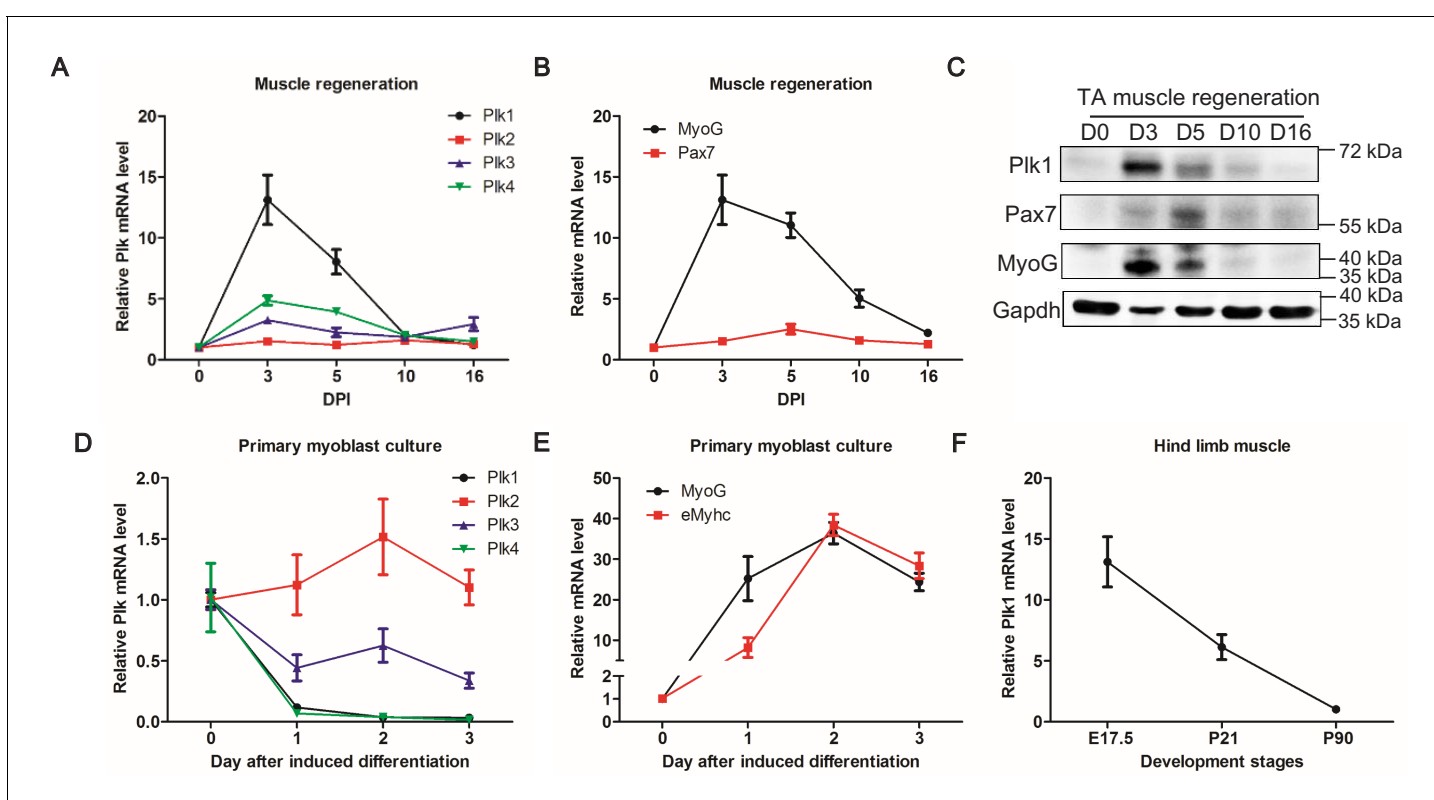

**Figure 1.** Expression patterns of *Plks* during muscle regeneration and differentiation. (**A–B**) Relative mRNA levels of *Plks* and myogenic factors *Pax7* and *MyoG* in TA muscles from mice (n = 4) at various timepoints after CTX induced injury, determined by qPCR, DPI: Days post injury; (**C**) Representative protein level of Plk1, Pax7 and Myog at various timepoints during muscle regeneration; (**D–E**) qPCR showing relative mRNA levels of *Plks* and myogenic differentiation markers (Myogenin and eMhyhc: embryonic myosin heavy chain) at various timepoints of primary myoblast differentiation (n = 3, biological samples); (**F**) qPCR analysis of *Plk1* expression in TA muscles from 17.5 day, 3 week and 3-month-old mice (n = 3, biological samples).
DOI: https://doi.org/10.7554/eLife.47097.003

Non-myogenic cells such as fibroadipogenic progenitors, endothelial cells, and infiltrating inflammatory cells are also present in the muscle during regeneration. To assure that the above-observed Plk1 dynamics are specific to MuSCs, we used an established antibody to detect Plk1 protein expression in MuSCs that were co-labeled with Pax7, an established marker of MuSCs (*Seale et al., 2000*). In the freshly isolated EDL myofibers (Day 0) carrying quiescent MuSCs, Plk1 immunofluorescence signal was undetectable in any Pax7$^+$ MuSCs (*Figure 2A*). After culture, Plk1 signal appeared in activated MuSCs that co-express MyoD (Day 1), and then was abundantly expressed in clusters of MuSCs progenies or myoblasts (Days 2–3), with positive signal only detected in MyoD$^+$ cells (*Figure 2A*, n > 200 cells), which marks activated MuSCs (*Olguin and Olwin, 2004*; *Zammit et al., 2004*). We also analyzed mRNA levels of *Plk1* in FACS-purified MuSCs isolated from non-injured and injured muscle tissues, representing quiescent and activated MuSCs, respectively. *Plk1* mRNA level was 15-fold higher in activated than in quiescent MuSCs (*Figure 2B*). Additionally, bFGF, a growth factor known to promote the proliferation of MuSCs, increased the level of *Plk1* in cultured myoblasts (*Figure 2C*). Collectively, these lines of evidence demonstrate that *Plk1* expression is dynamically regulated in MuSCs during their quiescence, activation, proliferation and differentiation.

## Loss of *Plk1* in myogenic progenitors leads to embryonic lethality

To assess the role of Plk1 in muscle development in vivo, we crossed the *Myod$^{Cre}$* with *Plk1$^{f/f}$* mice to generate the myoblast-specific *Plk1* knockout (*Myod$^{Cre}$::Plk1$^{f/f}$*, or Plk1$^{MKO}$) mice (*Figure 3A*). In this model, *Plk1* should be deleted in all muscle progenitors during development as *Myod$^{Cre}$* marks all embryonic myogenic cells (*Kanisicak et al., 2009*). After several intercrosses of heterozygous mice, no live Plk1$^{MKO}$ pups were obtained, indicating that myoblast-specific knockout of *Plk1* resulted in prenatal lethality. Therefore, embryos were harvested in utero at E16.5, several days before birth. From 69 embryos analyzed, we obtained 13 motionless embryos that were subsequently confirmed by PCR to be Plk1$^{MKO}$ (*Supplementary file 1*). Plk1$^{MKO}$ embryos were shorter in body length and transparent in appearance (*Figure 3B*). Histological sections through limbs revealed an absence of central-nucleated myofibers that were otherwise stained by eosin and embryonic myosin heavy chain (eMyHC) in the Plk1$^{MKO}$ embryos (*Figure 3C–D*). In contrast, eosin and eMyHC staining reveals the development of various groups of muscles in the WT control embryos (*Figure 3C–D*).

To determine the effect of Plk1$^{MKO}$ on myogenic progenitors, we labeled limb muscle sections with Pax7 antibody. Whereas numerous Pax7$^+$ myoblasts were detected in the WT control limb muscles, the Plk1$^{MKO}$ limb muscles were completely depleted of Pax7$^+$ myoblasts (*Figure 3E*), suggesting that Plk1 is necessary for the generation or maintenance of embryonic myoblasts in the limb muscles. In addition, no Ki67$^+$ cells were Pax7$^+$ in the Plk1$^{MKO}$ limb muscles (*Figure 3E*), suggesting that proliferative failure may be another driver of the depletion of myoblasts. These results indicate that loss of *Plk1* in myogenic progenitors blocks their proliferation and survival, leading to failure in skeletal muscle development and embryonic lethality.

## Deletion of *Plk1* impairs muscle regeneration in adult mice

The prenatal lethality of the Plk1$^{MKO}$ mice precludes the exploration Plk1 function in postnatal MuSCs and muscle regeneration. To circumvent embryonic lethality, we established tamoxifen (TMX)-inducible *Pax7$^{CreER}$::Plk1$^{f/f}$* (Plk1$^{PKO}$) mice to specifically delete *Plk1* in Pax7-expressing MuSCs upon TMX injection in adult mice (*Figure 4—figure supplement 1A*). WT control and Plk1$^{PKO}$ mice were intraperitoneally (IP) injected with TMX for five days followed by 14 days of chasing (*Figure 4—figure supplement 1A*). Knockout efficiency was shown by the lack Plk1 signal in MuSCs isolated from Plk1$^{PKO}$ myofibers but not in WT MuSCs after 72 hr of culture (*Figure 4—figure supplement 1B*). In the absence of injury, no morphological differences were observed between Plk1$^{PKO}$ and WT muscles based on H and E staining of tibialis anterior (TA) muscle cross-sections at Day 14 after TMX-induced deletion of *Plk1* (*Figure 4—figure supplement 1C–D*). Immunofluorescence staining of Pax7 showed that the number of MuSCs (Pax7$^+$/DAPI$^+$) in Plk1$^{PKO}$ mice was ~70% of that in WT mice, and no MyoG$^+$ differentiated cells were observed in either WT or KO muscles (*Figure 4—figure supplement 1E–G*). These results confirm the efficiency of the Plk1$^{PKO}$ conditional KO mouse model and demonstrate that the loss of *Plk1* do not lead to obvious morphological changes in uninjured muscles.

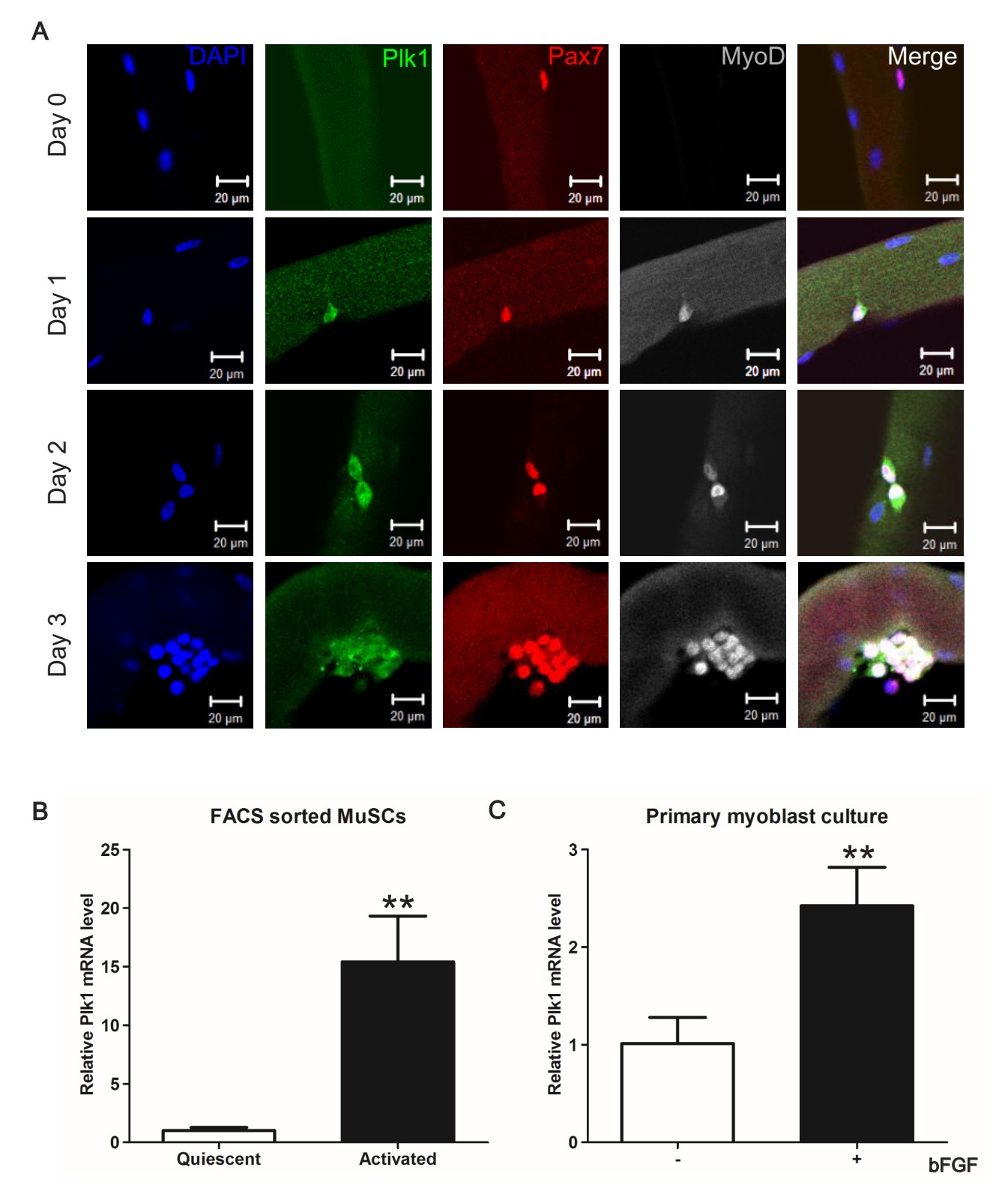

**Figure 2.** *Plk1* is specifically expressed in activated MuSCs. (**A**) Plk1 immunofluorescence in Pax7[+] MuSCs attached on freshly isolated EDL myofibers (Day 0) or after cultured for 1–3 days from three male wildtype mice (at least 10 fibers were collected at each time point, quantification of MyoD[+]Plk1[+] MuSCs were based on 200 cells on myofibers cultured for 72 hr), scale bar: 10 μm; (**B**) Relative levels of *Plk1* mRNA in quiescent and activated satellite
*Figure 2 continued on next page*

*Figure 2 continued*

cells, data represent mean ± s.e.m. (t-test: **p<0.01; n = 3, biological replicates); (C) Relative levels of *Plk1* mRNA in primary myoblast treated with (+) or without (-) 4 ng/ml bFGF for 24 hr, data represent mean ± s.e.m. (t-test: *p<0.05; n = 4, biological replicates).

DOI: https://doi.org/10.7554/eLife.47097.004

We then examined muscle regeneration of the Plk1$^{PKO}$ mice to determine the effect of *Plk1* KO on MuSCs. TA muscles from both Plk1$^{PKO}$ and WT mice were degenerated by CTX after TMX-induced *Plk1* deletion (*Figure 4A*). At 7 DPI, the masses of WT TA muscles were recovered to 70% of preinjury levels, whereas the TA muscle masses of Plk1$^{PKO}$ mice were only recovered by ~40% (*Figure 4B–C*). Histologically, the WT TA muscles were uniformly replaced by newly regenerated myofibers characterized by central-nucleated myofibers (*Figure 4D*). In contrast, TA muscles of Plk1$^{PKO}$ mice were devoid of newly regenerated myofibers (*Figure 4D*). Pax7$^+$ cells were diminished in Plk1$^{PKO}$ mice, together with an absence of any Dystrophin-expressing myofibers (*Figure 4E–F*). Furthermore, very few MyoG$^+$ cells and no eMyHC$^+$ myofibers were observed in Plk1$^{PKO}$ mice (*Figure 4—figure supplement 2A–C*). Compared with uniformly regenerated muscles in WT mice, Plk1$^{PKO}$ mice had no Pax7$^+$KI67$^+$ cells, but a compensatory increase in the number of Pax7$^-$KI67$^+$ cells (*Figure 4—figure supplement 2D–E*), indicating depletion of MuSCs and increased proliferation of non-myogenic cells. To further distinguish if Plk1 KO leads to regenerative delay or deficiency, we examined TA muscles at 21 DPI (*Figure 4—figure supplement 2F*), at which time the CTX-injured muscles were completely repaired, and the muscle weights were restored to the non-injured levels in the WT mice (*Figure 4—figure supplement 2G*). However, the TA muscle mass of the CTX-injured muscles was only 25% of the mass of the non-injured muscles in the Plk1$^{PKO}$ mice (*Figure 4—figure supplement 2G*). Newly regenerated myofibers (indicated by central-nucleation) were uniformly packed in the WT TA muscles (*Figure 4—figure supplement 2H*). In contrast, TA muscles of Plk1$^{PKO}$ mice were devoid of newly regenerated myofibers, and were infiltrated by Plin1 +adipocytes (*Figure 4—figure supplement 2H–I*) and F4/80$^+$ macrophages (*Figure 4—figure supplement 2I*). These results revealed that *Plk1* KO leads to ablation of MuSCs and severe regenerative failure.

To further dissect if Plk1 function in MuSCs and muscle regeneration depends on its kinase activity, we treated regenerating muscles with BI2536, a selective inhibitor of Plk1 activity (*Figure 4—figure supplement 3A*). BI2536 treatment (injection of 50 µl into the TA muscle at a final dosage of 0.4 µg/g body weight) significantly inhibited regeneration of CTX-injured muscles (*Figure 4—figure supplement 3B*). H and E staining of vehicle control and BI2536-treated muscle cross-sections revealed that BI2536-treated muscles had increased interstitial space filled with inflammatory infiltration and very few newly regenerated myofibers (*Figure 4—figure supplement 3C*). Consistent to *Plk1* KO results, BI2536 treatment did not have any effects on morphology of uninjured muscles (*Figure 4—figure supplement 3D*), suggesting Plk1 mainly functions to regulate proliferating MuSCs during regeneration. BI2536 treatment reduced the number of Pax7$^+$ cells by~60% in regenerating muscles, but not in uninjured muscle sections (*Figure 4—figure supplement 3E–F*). Similarly, BI2536 treatment did not affect the abundance of MuSCs in uninjured EDL myofibers, but significantly reduced the number of MuSCs on CTX-injured myofibers (*Figure 4—figure supplement 3G–H*). These data together suggest that genetic deletion or pharmacological inhibition of Plk1 similarly inhibits the proliferation of MuSCs and prevents muscle regeneration.

## *Plk1*-null MuSCs undergo cell cycle arrest at M Phase

To investigate how Plk1 regulate cell cycle progression of MuSCs, we synchronized WT myoblasts by double thymidine block and released them to allow synchronous cell cycle progression. Plk1 expression was significantly increased as cells enter G2 and peaked at metaphase, indicated by phosphohistone H3 (pHH3) labeling (*Figure 5A*). The cell-cycle-dependent expression pattern suggests a key role of Plk1 in regulating metaphase progression of MuSCs. To further determine cell cycle defects of Plk1 KO MuSCs, we analyzed proliferation of MuSCs attached on cultured EDL myofibers. After 3 days of culture, the WT MuSCs formed clusters of 6–8 myoblasts on the myofibers. However, MuSCs from Plk1$^{PKO}$ mice failed to complete the cytokinesis and were arrested at mitosis stage, evident from the dispersed distribution of nuclear Pax7 and MyoD signals surrounding metaphase and

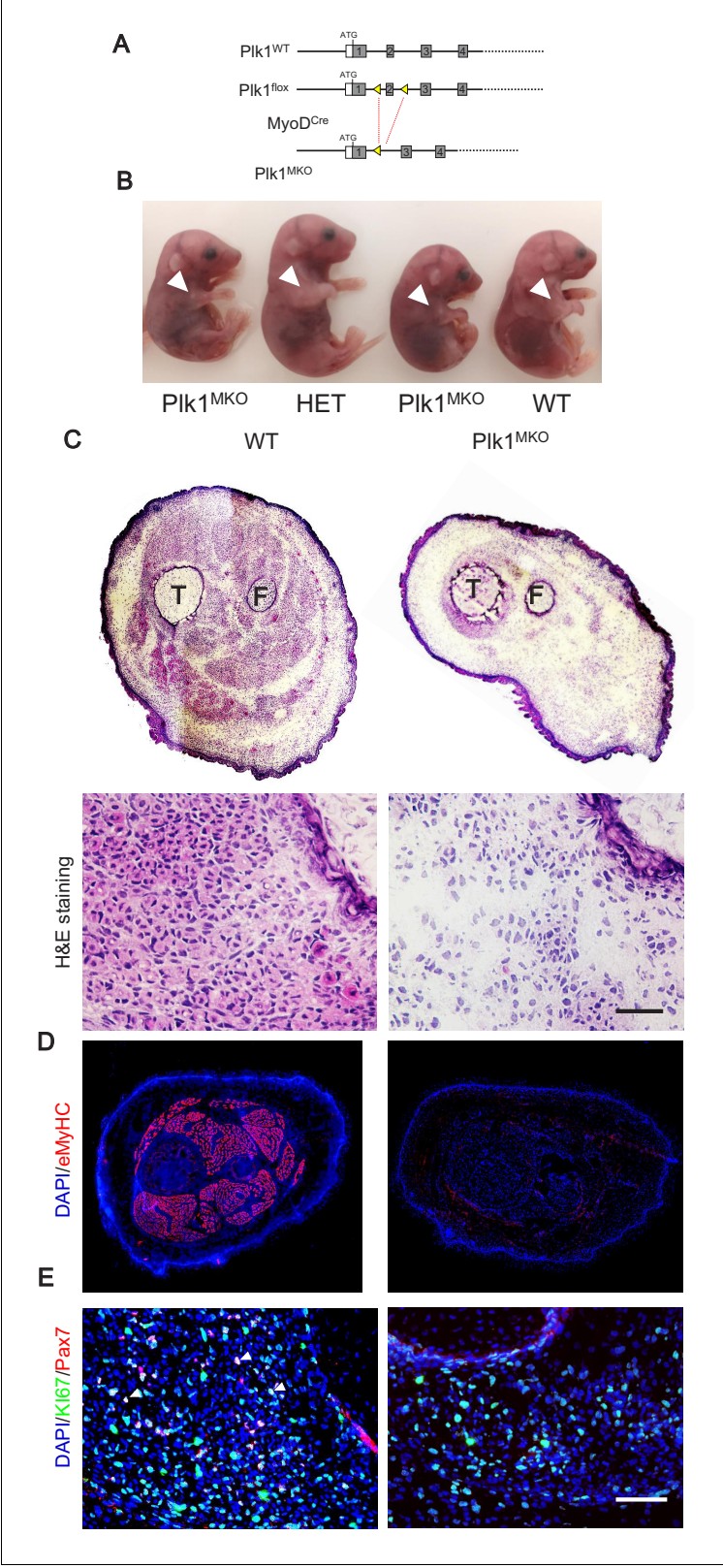

**Figure 3.** Loss of *Plk1* in myogenic progenitors leads to embryonic lethality. (**A**) Targeting strategy for myogenic progenitor specific deletion of Plk1, boxes represent exons and triangles represent LoxP; (**B**) Representative images of WT, heterozygous KO (Het) and Plk1^MKO embryos at stage E16.5, arrowheads points to the forelimbs, showing the; (**C**) H and E staining of E16.5 hindlimb cross-sections (upper panel) and magnified representative area (bottom panel, scale bar: 50 μm) showing lack of muscle fibers (labeled by eosin in pink) in Plk1^MKO embryos, T: Tibia, F: Fibula; (**D**)

*Figure 3 continued on next page*

Figure 3 continued

Immunofluorescence of eMyHC (marking myofibers) in E16.5 whole limb cross-sections; (E) Immunofluorescence of Pax7 and KI67 in limb cross-sections of embryos at E16.5, scale bar: 50 μm.

DOI: https://doi.org/10.7554/eLife.47097.005

telophase chromosomes (*Figure 5B*). The numbers of myoblasts were also significantly decreased on the Plk1[PKO] relative to WT myofibers (*Figure 5—figure supplement 1A*). We also isolated primary myoblast from Plk1[PKO] and used 4-hydroxy-tamoxifen (4-OHT) to induce acute *Plk1* deletion in proliferating myoblasts. The high efficiency of 4-OHT-induced *Plk1* deletion was verified by WB and DNA recombination analysis (*Figure 5D*). Compared to the control (Methanol) treatment in which the number of myoblasts increased exponentially, the numbers of 4-OHT-treated myoblast only slightly increased in the first 12 hr, followed by a gradual decrease (*Figure 5E*), suggestive of cell death. The numbers of myoblasts continually decreased when the myoblasts were cultured for more than 2 days (*Figure 5—figure supplement 1B*). Flow cytometry analyses revealed a significant increase of tetraploid cells 24 after 4-OHT induced KO of *Plk1* (*Figure 5F*). DAPI staining also demonstrated that over 60% of *Plk1*-null myoblast underwent a cell cycle arrest at M-phase with condensation of the chromatin and the disappearance of the nucleolus (*Figure 5G*).

The Plk1 inhibitor BI2536 was also used to treat cultured myofibers carrying MuSCs and primary myoblasts in vitro. When EDL myofibers were cultured for 3 days in the absence or presence of 100 nM BI2536, BI2536 treatment not only reduced the total number of myoblast clusters per myofiber by ~70%, but also reduced the number of cells per clusters, with no cluster containing more than four cells (*Figure 5—figure supplement 1C–D*). In contrast, 40% of the vehicle-treated myofibers contained clusters of more than four cells per cluster (*Figure 5—figure supplement 1D*). Consistently, BI2536 also suppressed the proliferation of primary myoblasts dose-dependently over various timepoints (*Figure 5—figure supplement 2A–B*). Nuclear staining with DAPI indicated that more than 40% of the BI2536-treated myoblasts were arrested at an undivided stage, resulting in tetraploid-like cells (*Figure 5—figure supplement 2C–D*), indicating an unsuccessful division of myoblasts by *Plk1* inhibition. Together, these data reveal a key role of Plk1 in cytokinesis and indicate that loss or inhibition of Plk1 leads to proliferative defect and cell cycle arrest at M-phase.

## *Plk1*-null MuSCs apoptose

We next questioned what's the fate of the *Plk1*-null myoblasts that were arrested at M-phase. To test this, we first checked the cell viability by antibody staining with cleaved Caspase-3 (C-Cas3), a marker for apoptosis (*Igney and Krammer, 2002*). As a positive control, 100 mM $H_2O_2$ was used to treat MuSCs on EDL myofibers cultured for 72 hr, which resulted in robust C-Cas3 signal (*Figure 6—figure supplement 1A*). Next, Plk1[PKO] EDL myofibers were cultured in vehicle control (Methanol) or 4-OHT to induce *Plk1* deletion. None of the methanol-treated myoblasts on myofibers were C-Cas3 positive, but all the 4-OHT treated MyoD[+] myoblasts were C-Cas3 positive at 48 and 72 hr after the treatment (*Figure 6A*). Morphologically, the 4-OHT-treated myoblasts were arrested in the cell cycle before the first division (*Figure 6A*). All activated myoblasts were C-Cas3 negative at 20 hr in both treatment group, indicative of a cell-cycle-dependent Cas3 activation. In situ terminal deoxynucleotidyl transferase dUTP nick-end labeling (TUNEL) also showed that MuSCs on EDL myofibers were all TUNEL[+] without finishing the first division after 42 hr culture (*Figure 6B*). Myoblasts cultured from Plk1[PKO] mice were also 100% C-Cas3[+] or TUNEL[+] when induced with 4-OHT following 2 days of culture, but control groups only had ~5% C-Cas3[+] or TUNEL[+] cells (*Figure 6C–F*). In addition, there were no presence of MyoG[+] differentiated myoblasts on EDL myofibers from Plk1[PKO] mice after 72 hr of culture (*Figure 6—figure supplement 1B*). Apoptosis of *Plk1* null myoblast was also confirmed by Annexin V-FITC Apoptosis Staining, which showed a dramatically increment of FITC/PI double positive cells 36 hr after 4-OHT induction when compared to the control group (*Figure 6—figure supplement 1C–D*). These results suggest that *Plk1* deletion induced loss of MuSCs due to apoptosis but not differentiation.

As Plk1 KO activates Cas3, which is necessary for myogenic differentiation (*Larsen et al., 2010*), we also examined the effect of Plk1 inhibition on differentiation of myoblasts. As deletion of *Plk1* in proliferating myoblasts leads to cell death and precludes analysis of cell differentiation, we used

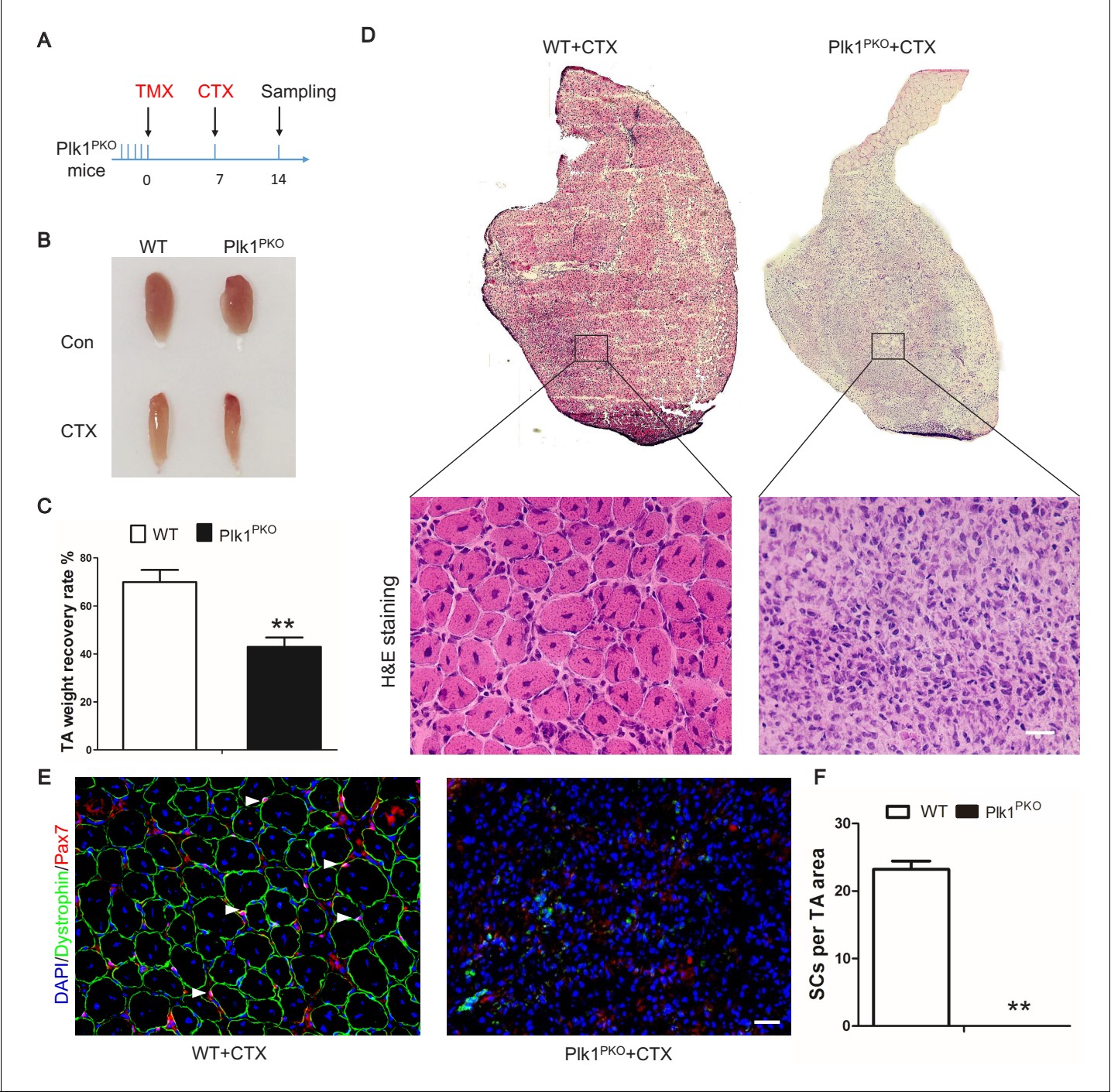

**Figure 4.** Plk1 deletion in MuSCs impairs muscle regeneration in vivo. (**A**) Experimental design for tamoxifen (TMX, four 8-week-old male mice from each group received a daily intraperitoneal injection for 5 consecutive days) induced deletion of *Plk1* in Pax7$^{CreER}$::*Plk1*$^{f/f}$ (Plk1$^{PKO}$) mice, following by cardiotoxin (CTX) injection to induce muscle degeneration and regeneration. Numbers indicate timing of TMX induction (5 consecutive days of injection following by 7 days of chasing), CTX injection and sample collection (7 days after injury); (**B**) Representative images of TA muscles in WT and Plk1$^{PKO}$ mice; (**C**) TA muscle weight recovery at 7 days after CTX injury, data represent mean ± s.e.m. (t-test: **p<0.01; n = 4, biological replicates, mice were males at around 11 weeks old when sampled); (**D**) H and E staining of TA muscle cross-sections (upper panel) and magnified representative areas at 7 days after CTX injury (bottom panels, scale bar: 20 μm); (**E**) Immunofluorescence of Pax7 and Dystrophin (to outline myofibers) in TA muscle cross-sections at 7 days after CTX injury, scale bar: 20 μm; (**F**) The average number of MuSCs per microscopic area, data represent mean ± s.e.m. (t-test: **p<0.01; n = 4, biological replicates, all MuSCs from five radom areas were counted for each mouse).

DOI: https://doi.org/10.7554/eLife.47097.006

*Figure 4 continued on next page*

*Figure 4 continued*

The following figure supplements are available for figure 4:

**Figure supplement 1.** Plk1 deletion in MuSCs have minimal impact on resting muscle.

DOI: https://doi.org/10.7554/eLife.47097.007

**Figure supplement 2.** Plk1 deletion in MuSCs impairs muscle regeneration in vivo.

DOI: https://doi.org/10.7554/eLife.47097.008

**Figure supplement 3.** Plk1 inhibition by BI2623 impairs muscle regeneration.

DOI: https://doi.org/10.7554/eLife.47097.009

BI2536 to inhibit Plk1 activity in myoblasts. Interestingly, myosin heavy chain (marked by MF20), markers of myogenic differentiation, was observed in 15% of myoblasts treated with BI2536, but rarely observed in control myoblasts cultured in growth media without BI2536 (*Figure 6—figure supplement 2A–B*). This suggests that Plk1 inhibition promotes premature differentiation of myoblasts. When BI2536 was added to myoblasts at the onset of serum withdrawal-induced myogenic differentiation, the fusion index at 48 hr was increased by 30% compared to vehicle treatment controls (*Figure 6—figure supplement 2C–D*). The protein and mRNA levels of MF20 were increased by Plk1 inhibition under both growth and differentiation conditions (*Figure 6—figure supplement 2E–F*). These results demonstrate that pharmacological inhibition of Plk1 in primary myoblasts promote their differentiation.

## DNA damage response and p53 are induced in *Plk1*-null MuSCs

To understand how Plk1 deletion and inhibition leads to the apoptosis of myoblasts, we first examined DNA damage response (DDR), revealed by the presence of phosphorylated histone 2A family member X (γH2AX) (*Paull et al., 2000*). We found that *Plk1* null MuSCs (labeled by Cav1) that were arrested at undivided stage had very strong γH2AX signal (*Figure 7A*). This is in sharp contrast to control cells that contained very few small puncta of γH2AX signal (*Figure 7A*). Similarly,~80% of cultured *Plk1* KO primary myoblasts were strongly γH2AX$^+$ at 2 days after 4-OHT induction, whereas only ~23% control WT myoblast exhibited weak γH2AX signal (*Figure 7B–C*). We also analyzed DNA fragmentation of BI2536 treated myoblasts using single-cell gel electrophoresis assay, revealing that 70% of myoblasts accumulated DNA damage after BI2536 treatment (*Figure 7D*). Among these, 55% of myoblasts were scored as moderate damage (Classes 1–3) and 15% were scored as maximal damage (Class 4) (*Figure 7E*). In contrast, 95% control myoblasts were fragmentation-free and only 5% control myoblasts displayed moderate DNA damage (*Figure 7E*). We also synchronized cell cycle with double thymidine block and release (*Figure 7—figure supplement 1A*). The *Plk1* KO (4-OHT treated) myoblasts exhibited more abundant γH2AX signal than control myoblasts at 8 hr after the release, but no differences in TUNEL signal were observed between the two groups (*Figure 7—figure supplement 1B*), suggesting DNA damage precedes apoptosis. At 12 hr after release, TUNEL$^+$ myoblasts were significantly increased in 4-OHT treated groups (~80%) and all the 4-OHT-treated myoblast were TUNEL positive at 16 hr and 24 hr after release. Consistently, we also found that γH2AX signal was increased by approximately fourfold in BI2536-treated myoblasts compared to control myoblasts (*Figure 7—figure supplement 1C–D*). These results are in agreement with the pivotal role of Plk1 in DNA damage checkpoint regulation (*Takaki et al., 2008*), and demonstrate that deletion of Plk1 in MuSCs accumulates DNA damage response and leads to apoptosis.

Finally, we explored the molecular mechanism mediating the defective DDR and apoptosis of *Plk1*-null MuSCs. The tumor suppressor p53, known to play a pivotal role in DDR and cell survival, was reported to be upregulated by Plk1 inhibition in cancer cells (*Liu and Erikson, 2003*). WB analysis showed that p53 protein level was massively elevated in *Plk1* KO myoblasts compared to control myoblasts (*Figure 7F*). In contrast, *Plk1* KO did not affect the level of Phospho-ATR (*Figure 7F*), which has been reported to play a role in G2 checkpoint (*Cimprich and Cortez, 2008*). To examine if p53 upregulation is responsible for cell apoptosis, we used Pifithrin-α (2 nM, Sigma) to inhibit p53 in control and Plk1 KO myoblasts. We found that 4-OHT-induced deletion of *Plk1* led to more than 80% of apoptotic cells (C-Caspase three positive) within 36 hr, and inhibition of p53 significantly reduced apoptosis of *Plk1*-null myoblasts (*Figure 7G,H*). However, inhibition of ATR failed to rescue the apoptosis of *Plk1*-null myoblasts (*Figure 7H*), suggesting that apoptosis occurred prior to the

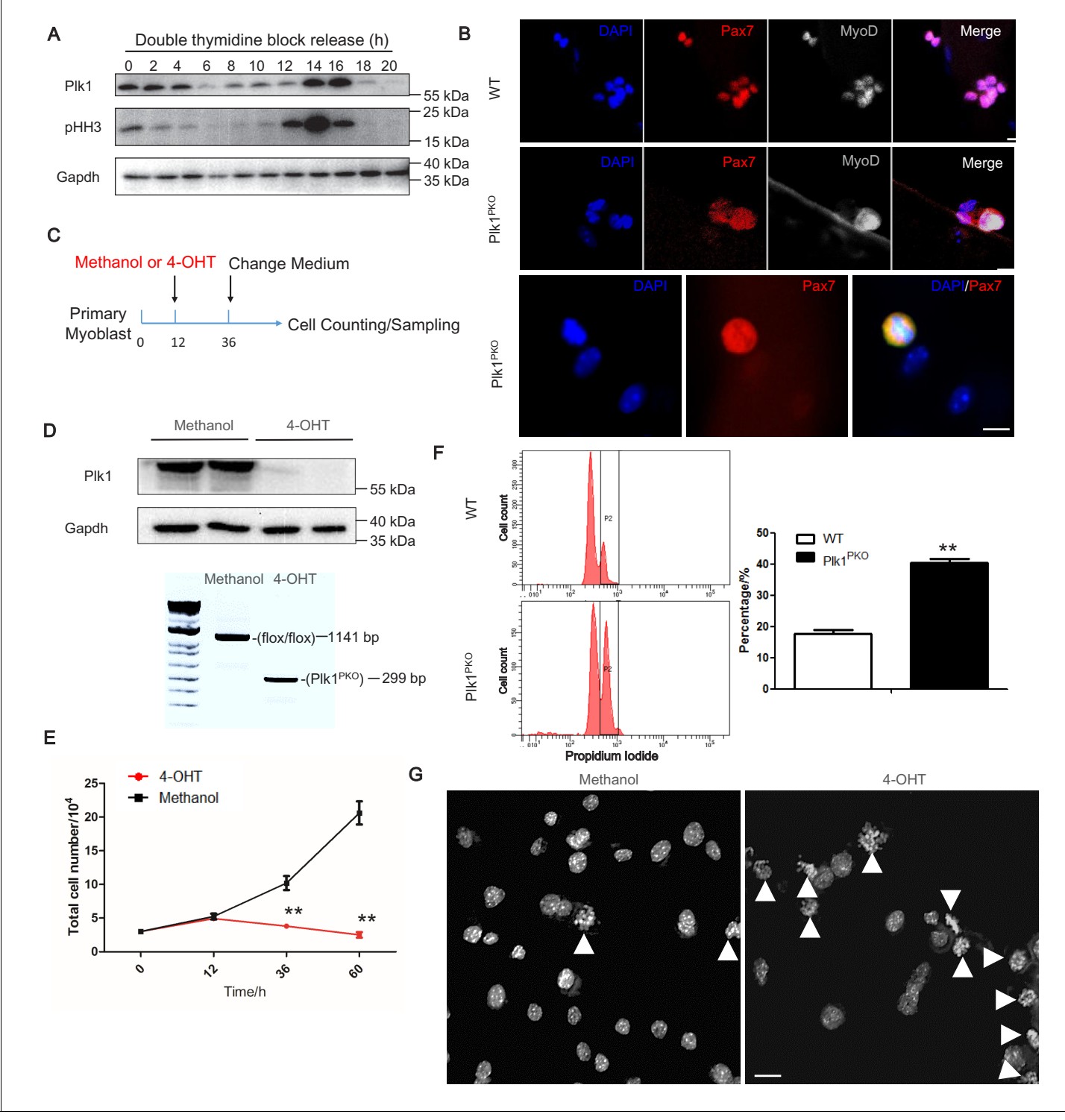

**Figure 5.** Plk1 deletion in MuSCs leads to cell cycle arrest. (**A**) Western blot showing relative protein levels of Plk1 and cell cycle marker pHH3 at different time points after WT myoblast were synchronized by double thymidine block (DTB) protocol and released; (**B**) Immunofluorescence of Pax7 and MyoD in MuSCs on cultured EDL myofibers (72 hr) isolated from WT and Plk1[PKO] mice 7 days after TMX induced deletion of *Plk1*, scale bar: 10 μm, (at least 10 fibers were collected from four individual 11-week-old male mice); (**C**) Schematics showing 4-hydroxy-tamoxifen (4-OHT) induced deletion of *Plk1* in primary myoblasts isolated from Plk1[PKO] mice (Primary myoblasts were isolated from four 6-week-old male mice, and frozen at −80°C and stored for the following experiments), Methanol treatment is the vehicle control; (**D**) Western blot showing effective knockout of Plk1 after 4-OHT induction (upper panel) and PCR analysis of genomic DNA showing the DNA recombination 4-OHT induction; (**E**) Quantification of the average numbers of
*Figure 5 continued on next page*

Figure 5 continued

myoblast per well, data represent mean ± s.e.m. (t-test: **p<0.01; n = 4, biological replicates); (F) Flow cytometry analysis of methanol or 4-OHT treated myoblast with propidium iodide staining to mark DNA content (left panel), and quantification of percentage of myoblasts containing double-DNA content (right panel), data represent mean ± s.e.m. (t-test: **p<0.01; n = 3, biological replicates); (G) DAPI staining of myoblasts at 30 hr after methanol or 4-OHT induction to reveal DNA morphology, scale bar: 10 μm, white arrowheads indicated cells containing unsegregated chromosomes.

DOI: https://doi.org/10.7554/eLife.47097.010

The following figure supplements are available for figure 5:

**Figure supplement 1.** Plk1 deletion in MuSCs leads to cell cycle arrest.

DOI: https://doi.org/10.7554/eLife.47097.011

**Figure supplement 2.** Plk1 inhibition in myoblast leads to cell cycle arrest.

DOI: https://doi.org/10.7554/eLife.47097.012

G2-checkpoint. Interestingly, inhibition of p53 or ATR/ATM alone in WT myoblasts increased their apoptosis (*Figure 7H*), consistent with the notion that p53 and ATR promotes cell survival in the absence of DNA damage (*Reinhardt et al., 2007*; *Sakaguchi et al., 1998*). Altogether, these results point to the upregulation of p53 as a key driver of cell cycle arrest and apoptosis in the *Plk1*-null MuSCs.

## Discussion

The expansion of MuSCs plays a crucial role for muscle development and repair following injury and a variety of signal pathways are involved in the regulation of myoblast proliferation and expansion (*Conboy and Rando, 2002*; *Jones et al., 2001*; *Jones et al., 2005*; *Perdiguero et al., 2007*). Although Plk1 has been shown to play an indispensable role in cell cycle and mitotic progression of non-myogenic cell types especially cancer cells (*Barr et al., 2004*), its function in MuSCs and myogenesis has not been investigated. Here, we identify an irreplaceable role of Plk1 in controlling MuSC proliferation. Specifically, we found that *Plk1*-deficient MuSCs lose their proliferative capacity, accumulate marks of DNA damage response and undergo apoptosis without completing mitosis, leading to failure of muscle development and defective muscle regeneration in response to injury.

PLK1 is highly expressed in various human tumors and proliferating cells during embryonic development (*Lu et al., 2008*; *van Vugt and Medema, 2005*). Our findings extend this feature to the adult stem cells and demonstrate that Plk1 activity synchronizes with the proliferative ability of MuSCs. Previous studies have reported that constitutive deletion of *Plk1* leads to embryonic lethality at morula stage (E3.5) due to mitotic aberrancies (*Wachowicz et al., 2016*). Similar mitotic defects were also observed in our myogenic progenitor-specific *Plk1* KO mice, which died prenatally due to failure in muscle development. Compared to our model, *Pax7$^{-/-}$* mice exhibit progressive loss of the MuSC lineage with reduced muscle size, myonuclei number and myofiber diameters, leading to poor viability and early death within the first 3 weeks of life (*Seale et al., 2000*). *Myf5* and *MyoD* double knockout mice, but not the *Myf5* or *MyoD* single knockout mice, show complete lack of skeletal muscle formation (*Rudnicki et al., 1992*; *Rudnicki et al., 1993*). *Myogenin*-null mice die after birth from severe and global muscle deficiency (*Hasty et al., 1993*). Our myogenic progenitor specific *Plk1* KO mice die earlier than these KO mice lacking one of the key myogenic transcription factors, demonstrating an indispensable role of Plk1 in embryonic muscle development.

Activation of MuSCs is a crucial step for muscle regeneration. Evidence shows that a lack of MuSCs contributes to defective regeneration and muscle weakness (*Relaix and Zammit, 2012*). Indeed, previous studies have demonstrated that blockage of cell proliferation by colchicine treatment (*Pietsch, 1961*) or irradiation (*Quinlan et al., 1995*) drastically reduces muscle regenerative capacity. In addition, MuSCs function is also controlled by various signals in the local microenvironment, called stem cell niche (*Kuang et al., 2008*). This niche consists of various cells and extracellular factors that function to regulate MuSCs during muscle regeneration (*Kuang et al., 2008*). Our study identifies Plk1 as an intrinsic regulator of MuSCs, but what extrinsic factors regulate the dynamic expression of *Plk1* remains to be elucidated. It is reported that ~ 50% reduction of MuSCs induced by *Pten* deletion could sufficiently maintain muscle regeneration, and muscle regeneration fails only when the SC population drops to 10% or less (*Yue et al., 2017*). In our model, MuSCs-specific *Plk1* deletion only leads to ~30% reduction in MuSCs, yet the remainder 70% MuSCs fail to regenerate

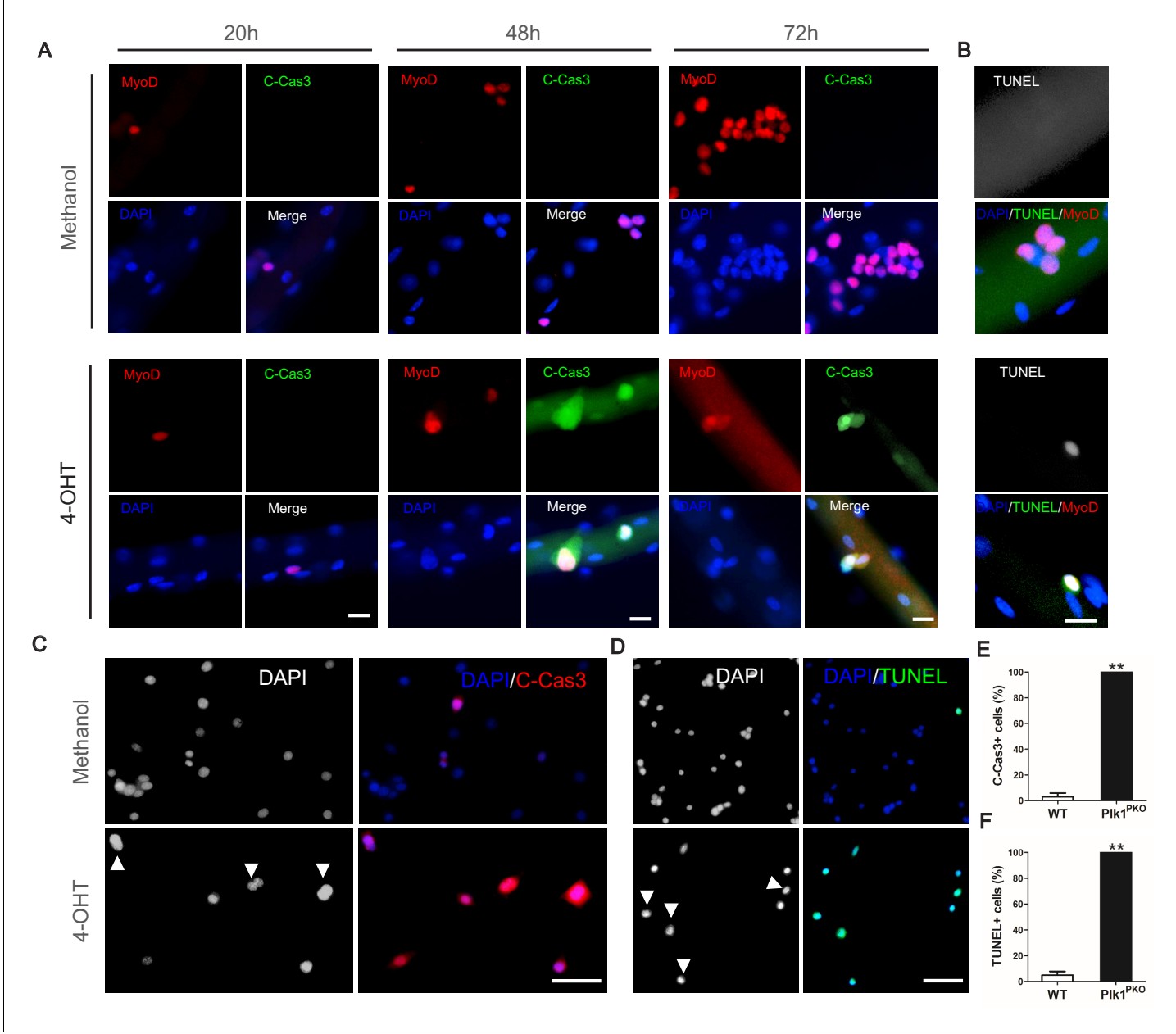

**Figure 6.** *Plk1* deletion leads to apoptosis of MuSCs. (**A**) Representative images of MyoD (red) and cleaved Caspase-3 (green) staining in MuSCs located on EDL myofibers cultured for 20, 48 and 72 hr, myofibers were isolated from Plk1PKO mice and treated with 4-OHT to induce deletion of Plk1 at time 0, Methanol is the vehicle control, scale bar: 10 μm, at least 10 cells from 10 single myofibers per mouse (n = 3 mice) were counted; (**B**) Immunofluorescence of Myod (red) and TUNEL (green) in MuSCs on EDL myofibers cultured for 42 hr, scale bar, 10 μm, at least 10 cells from 10 single myofibers per mouse (n = 3 mice) were counted; (**C–D**) Representative images of cleaved Caspase-3 (**C**) and TUNEL (**D**) staining in primary myoblasts at 48 hr after methanol or 4-OHT treatment, nuclei were counterstained with DAPI, arrowheads indicate cells arrested at M-phase, scale bar: 50 μm; (**E–F**) Quantification of percentages of Caspase-3+ (**E**) and TUNEL+ (**F**) cells, data represent mean ± s.e.m. (t-test: **p<0.01; n = 3, biological replicates (mice), three replicates/mouse, 25–50 myoblasts/replicate were analyzed).
DOI: https://doi.org/10.7554/eLife.47097.013

The following figure supplements are available for figure 6:

**Figure supplement 1.** Deletion of Plk1-null MuSCs is due to apoptosis but not differentiation.
DOI: https://doi.org/10.7554/eLife.47097.014

**Figure supplement 2.** Inhibition of Plk1 leads to premature differentiation of primary myoblasts.
DOI: https://doi.org/10.7554/eLife.47097.015

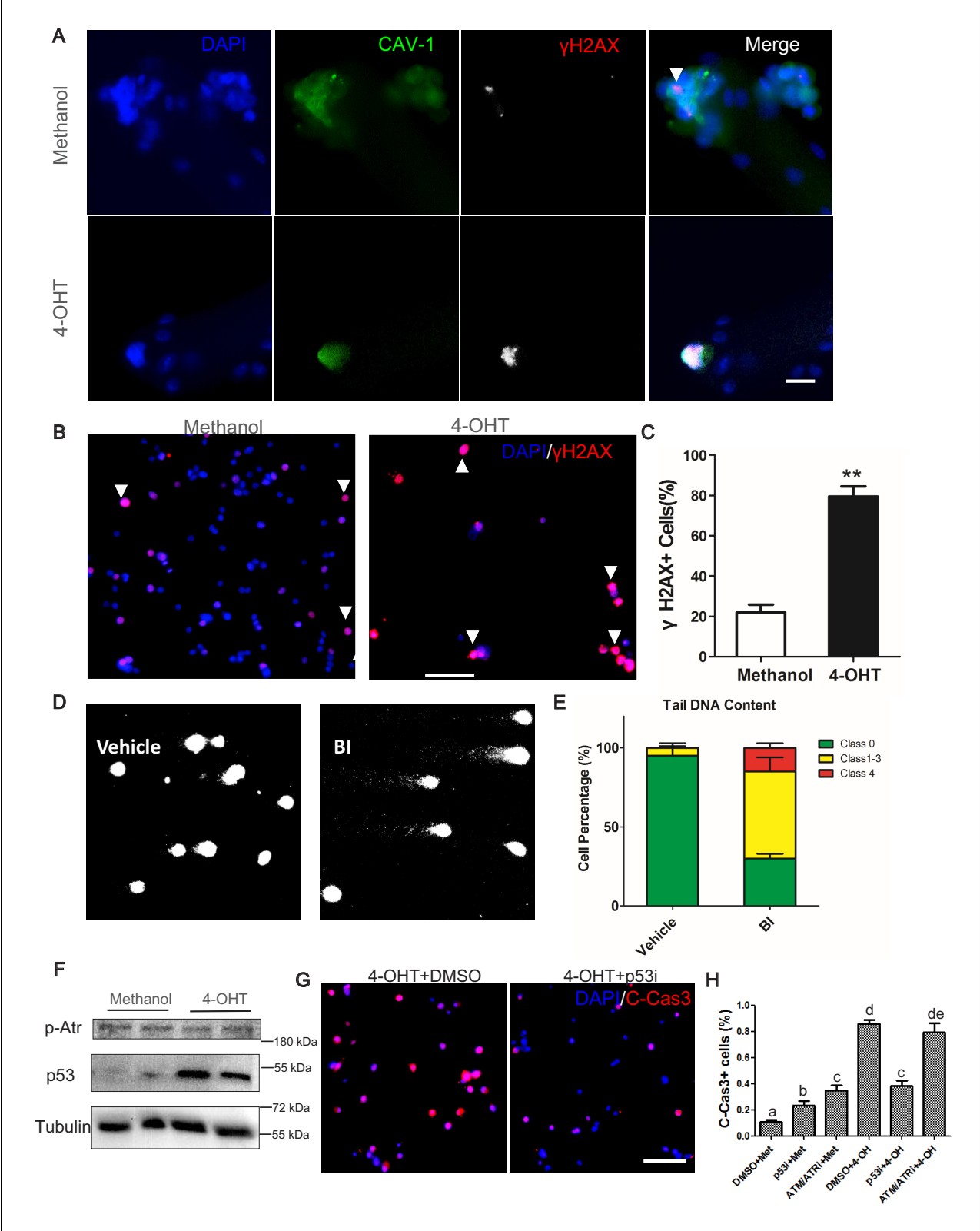

**Figure 7.** *Plk1*-null MuSCs excessively accumulate markers of DNA damage response and undergo p53-mediated apoptosis. (A) Immunofluorescence of γH2AX (red) and Caveolin-1 (green) in MuSCs on EDL myofibers cultured for 72 hr. Myofibers were isolated from Plk1^PKO mice and treated with 4-OHT to induced *Plk1* deletion (methanol: vehicle control), scale bar: 10 µm; (B) Representative images of γH2AX staining in primary myoblast isolated from Plk1^PKO mice, treated with 4-OHT or methanol, and cultured for 48 hr, nuclei were counterstained with DAPI, scale bar: 50 µm; (C) Quantification

*Figure 7 continued on next page*

*Figure 7 continued*

of percentages γH2AX+ myoblasts after treatment shown in B, data represent mean ± s.e.m. (t-test: **p<0.01; n = 3, biological replicates(mice), three replicates/mouse, 25–50 myoblasts/replicate were analyzed); (D) Representative images of single-cell gel electrophoresis under alkaline conditions in primary myoblasts 24 hr after BI2536 treatment; (E) Percentage of myoblasts with different levels of DNA damage, presented as no damage, moderate damage (classes 1–3) and maximal damage (class 4), data represent mean ± s.e.m. (t-test: *p<0.05; n = 3, biological replicates (mice), three replicates/mouse, 25–50 myoblasts/replicate were analyzed); (F) Western blot showing relative levels of p53 and Phospho-ATR in Plk1PKO primary myoblast at 24 hr after methanol or 4-OHT treatment; (G) Representative images of cleaved Caspase-3 (red) staining in primary myoblast 24 hr after 4-OHT together with DMSO (left panel) or p53 inhibitor (p53i) treatment (right panel), nuclei were counterstained with DAPI, scale bar: 50 μm, (H) Quantification of C-Cas3+ apoptotic cells at 24 hr after methanol or 4-OHT together with p53 inhibitor or ATM/ATR inhibitor treatment, data represent mean ± s.e.m. (t-test: p<0.05; n = 3, biological replicates, 25–50 myoblasts from three replicates from three individual mice were counted for quantification).
DOI: https://doi.org/10.7554/eLife.47097.016

The following figure supplement is available for figure 7:

**Figure supplement 1.** *Plk1*-null MuSCs undergo DNA damage response induced apoptosis.
DOI: https://doi.org/10.7554/eLife.47097.017

the injured muscle after CTX injury due to cell cycle arrest. This leads to infiltration of fibro-adipogenic progenitors (FAPs), macrophages and adipocytes that occupy the muscle after CTX-induced regeneration. These results highlight the absolute requirement of Plk1 in cell cycle progression of *Pax7*-expressing MuSCs during regenerative myogenesis.

PLKs are key regulators of many cell-cycle-related events, including chromosome segregation, centrosome maturation, bipolar spindle formation, regulation of anaphase-promoting complex, and execution of cytokinesis (*Barr et al., 2004*). Conditional knockout of *Plk1* leads to defective polyploidization and cell death in megakaryocytes (*Trakala et al., 2015*). *Plk1* inhibition also prevented pancreatic β cell proliferation and G2/M cell-cycle phase progression in a dose-dependent manner (*Shirakawa et al., 2017*). In addition to proliferative defects, *Plk1*-null MuSCs also undergo apoptosis. This observation is consistent with previous reports that *Plk1* depletion induces apoptosis in cancer cells (*Liu and Erikson, 2003*). During normal mitosis processes, if DNA damages are detected a signaling cascade will be initiated to enforce cell cycle arrest (checkpoint activation), followed by DNA repair process. If DNA repair fails or if excessive DNA lesions are accumulated, an apoptosis process will be triggered. Phosphorylated H2AX could form nuclear foci within 1 min at the sites of DNA double-strand breaks and thus represents a sensitive marker of DNA damages (*Paull et al., 2000*). Here, we reported that after *Plk1* deletion, most MuSCs exhibit robust γH2AX signal, suggesting that a failure of DNA repair may underpin the observed cell death.

Previous work has also shown that PLK1 plays a crucial role during recovery from G$_2$ DNA damage checkpoint through targeting multiple factors such as ATR/Chk1, ATM/Chk2 and p53 pathways (*Bassermann et al., 2008*; *Li et al., 2017*). We show that pharmacological inhibition of p53 or ATM/ATR significantly increases the apoptosis of wildtype myoblasts, potentially due to inhibition of p53 and ATM-dependent DNA damage response. The Plk1-null cells also underwent apoptosis, manifested by markers of activated Caspase-3, DNA damage response (γH2AX), DNA fragmentation and TUNNEL labeling. Previous studies have shown that the DNA damage response and activation of Caspase-3 are key steps in myogenic differentiation (*Burgon and Megeney, 2018*; *Fernando et al., 2002*; *Fortini et al., 2012*; *Larsen et al., 2010*). In agreement with this concept, we also found that pharmacological inhibitions of Plk1 activates Caspase-3 and promotes myogenic differentiation. However, the *Plk1*-null myoblasts fail to express the myogenic differentiation marker myogenin despite activation of Caspase-3 (*Figure 4—figure supplement 1F* and *Figure 6—figure supplement 1B*), due to the cell cycle blockage and apoptosis prior to terminal differentiation. These results demonstrate that cell cycle-dependent dynamic regulation Plk1 is key for MuSC proliferation and differentiation. PLK1 knockout in tumor cells induces DNA damage and causes p53 activation (*Li et al., 2017*; *Liu and Erikson, 2003*), but activated p53 could also transactivate proapoptotic genes, leading to cell death (*Liu and Erikson, 2003*). This explains why p53 inhibition partially rescues apoptosis of *Plk1*-null myoblasts that express high levels of p53. Our results demonstrate that a proper level of p53 is critical for MuSC homeostasis. Indeed, it has been reported that regeneration-induced loss of quiescence in *p53*-deficient MuSCs results in tumor formation (*Preussner et al., 2018*). Similarly, lower levels of p53 is observed in aged MuSCs that exhibit a high frequency of cell death in the transient expansion phase of muscle regeneration (*Liu et al., 2018*).

PLK1 is highly expressed in several cancer types, and thus represents a druggable target in cancer therapeutics. BI2536, one of the most effective Plk1 inhibitors, induces apoptosis of rhabdomyosarcoma cells when synergistically used with microtubule-destabilizing drugs, but a low dose of BI2536 (7 nM) has no effect on C2C12 myoblasts (*Hugle et al., 2015*). Consistently, we report that low dose of BI2536 (1–10 nM) have no effect on the proliferation of MuSCs-derived primary myoblasts. However, BI2526 treatment profoundly inhibits MuSCs function during muscle regeneration, cautioning the potential side effect of PLK1 inhibition in skeletal muscle homeostasis and cancer cachexia. Impaired regenerative ability has been well-established in aged muscle owing in part to decreased number and functionality of MuSCs (*Jang et al., 2011*), including defective dividing capability (*Liu et al., 2018*) and quiescent maintenance failure (*Chakkalakal et al., 2012*). Similar aging features were observed in our *Plk1* null or BI2536 treated MuSCs, suggesting that increasing Plk1 activity may preserve functions of MuSCs in aged muscle.

# Materials and methods

## Key resources table

| Reagent type (species) or resource | Designation | Source or reference | Identifiers | Additional information |
|---|---|---|---|---|
| Genetic reagent (*M. musculus*) | *Plk1*[flox] | Pubmed | PMID: 27417127 | Dr. Guillermo de Cárcer (Spanish National Cancer Research Centre) |
| Genetic reagent (*M. musculus*) | *Myod*[cre] | Jackson Laboratory | Stock #: 014140 RRID:IMSR_JAX:014140 | |
| Genetic reagent (*M. musculus*) | *Pax7*[creER] | Jackson Laboratory | Stock #: 012476 RRID:IMSR_JAX:012476 | |
| Antibody | Pax7(PAX7) mouse monoclonal | Developmental Studies Hybridoma Bank | Cat# pax7 RRID:AB_528428 | IHC (1:10) |
| Antibody | MyoG mouse monoclonal | Developmental Studies Hybridoma Bank | Cat# PCRP-MYOG-1C5 RRID:AB_2722260 | IHC (1:500) WB (1:1000) |
| Antibody | MF20 mouse monoclonal | Developmental Studies Hybridoma Bank | Cat# MF 20 RRID:AB_2147781 | IHC (1:50) WB (1:200) |
| Antibody | eMyHC mouse monoclonal | Developmental Studies Hybridoma Bank | Cat# F1.652 RRID:AB_528358 | IHC (1:100) |
| Antibody | Plk1 rabbit polyclonal | Cell Signaling | Cat# 4535 RRID:AB_2252687 | IHC (1:500) WB (1:2000) |
| Antibody | MyoD mouse monoclonal | Santa Cruz Biotechnology | Cat# sc-377460 | IHC (1:300) |
| Antibody | Dystrophin rabbit polyclonal | Abcam | Cat# ab15277 RRID:AB_301813 | IHC (1:1000) |
| Antibody | Ki67 rabbit polyclonal | Abcam | Cat# ab15580 RRID:AB_443209 | IHC (1:1000) |
| Antibody | Phospho-Histone H3 rabbit polyclonal | Cell Signaling Technology | Cat# 9701 RRID:AB_331535 | WB (1:1000) |
| Antibody | Phospho-ATR rabbit monoclonal | Cell Signaling Technology | Cat# 30632 RRID:AB_2798992 | WB (1:1000) |
| Antibody | p53 rabbit polyclonal | Cell Signaling Technology | Cat# 9282 RRID:AB_331476 | WB (1:1000) |
| Antibody | Cleaved Caspase-3 rabbit polyclonal | Cell Signaling Technology | Cat# 9661 RRID:AB_2341188 | IHC (1:500) |

*Continued on next page*

*Continued*

| Reagent type (species) or resource | Designation | Source or reference | Identifiers | Additional information |
|---|---|---|---|---|
| Antibody | Anti-H2A.X, phospho mouse monoclonal | Abcam | Cat# ab26350 RRID:AB_470861 | IHC (1:1000) |
| Antibody | 488 goat polyclonal antimouse IgG1 | Thermo Fisher Scientific | Cat# A-21121 RRID:AB_141514 | IHC (1:1000) |
| Antibody | 488 goat polyclonal antirabbit IgG | Thermo Fisher Scientific | Cat# A-11034 RRID:AB_2576217 | IHC (1:1000) |
| Antibody | 568 goat polyclonal antimouse IgG1 | Thermo Fisher Scientific | Cat# A-21124 RRID:AB_2535766 | IHC (1:1000) |
| Antibody | 488 goat polyclonal antirabbit IgG | Thermo Fisher Scientific | Cat# A-21244 RRID:AB_2535812 | IHC (1:1000) |
| Antibody | HRP-conjugated goat polyclonal anti-rabbit IgG | Jackson Immuno Research Labs | Cat# 111-035-003 RRID:AB_2313567 | WB (1:10000) |
| Antibody | HRP-conjugated goat polyclonal anti-mouse IgG | Jackson Immuno Research Labs | Cat# 115-035-003 RRID:AB_10015289 | WB (1:10000) |
| Chemical compound, drug | BI2536 | Selleckchem | Cat# S1109 | |
| Chemical compound, drug | Pifithrin-α | Sigma | Cat# P4359 | |
| Chemical compound, drug | CGK733 | Sigma | Cat# C9867 | |
| Chemical compound, drug | propidium iodide | Sigma | Cat# P4170 | |
| Chemical compound, drug | Annexin V | Invitrogen | Cat# V13241 | |

## Mice and animal care

*Myod$^{Cre}$* (#014140), *Pax7$^{CreER}$* (#012476) mice were obtained from Jackson Laboratory and housed in the animal facility with free access to water and standard rodent chow. *Plk1$^{f/f}$* mice was a gift from Dr. Cárcer from Spanish National Cancer Research Centre (CNIO), Madrid, Spain (*Wachowicz et al., 2016*). Mice were genotyped by PCR of ear DNA using primers listed in *Supplementary file 2*. The genotypes of experimental KO and associated control animals are as follows: Plk1$^{PKO}$ (*Pax7$^{CreER}$:: Plk1$^{f/f}$*) and wild type (*Plk1$^{f/f}$*), Plk1$^{MKO}$ (*Myod$^{Cre}$::Plk1$^{f/f}$*) and wild type (*Plk1$^{f/f}$*). Mouse maintenance and experimental use were performed according to protocols approved by the Purdue Animal Care and Use Committee.

## Muscle injury and regeneration

Muscle regeneration was induced by injections of cardiotoxin (CTX, sigma) into the tibialis anterior (TA) muscles of 8–12 week-old male mice. Mice were anesthetized using a ketamine-xylazine cocktail, and 50 µl saline or 50 µl of 10 µM CTX was injected into TA muscles in the absence or presence of 10 µg BI2536. Muscles were then harvested at different days post-injection to assess the completion of regeneration, repair and gene expression. Tamoxifen (Calbiochem) was prepared in corn oil at a concentration of 10 mg ml$^{-1}$, and experimental and control mice were injected intraperitoneally at 2 mg per day per 20 g body weight for 5 days to induce Cre-mediated deletion.

## Isolation, culture and differentiation of primary myoblast

Primary myoblasts were isolated from hind limb skeletal muscle of 6-week-old male mice as previously described (*Yue et al., 2017*). Muscle tissues were minced and digested in type II collagenase

and Dispase B mixture (Roche). Digested cells were harvested and cultured in growth media, F-10 Ham's medium (Thermo Fisher Scientific) supplemented with 20% fetal bovine serum (FBS, Atlanta), 4 ng/ml basic fibroblast growth factor (Thermo Fisher Scientific) and 1% penicillin-streptomycin (Thermo Fisher Scientific) on collagen-coated dishes. Primary myoblasts were isolated and purified after 2–3 times of pre-plate. For in vitro genetic deletion, 4-OHT (0.4 µM, Calbiochem) was added in culture medium for 1 day to induce Cre-mediated deletion. P53 (Pifithrin-α, Sigma) and ATM/ATR (CGK733, Sigma) inhibitor were used according to the manufactory's protocol and treated the myoblast together with 4-OHT or Methanol. Muscle differentiation was induced using 80% confluence of isolated primary myoblasts in Dulbecco's Modified Eagle Medium (DMEM, Sigma,) supplemented with 2% horse serum (Sigma). Differentiated cells were kept in differentiation media for further analysis.

## Flow cytometry analysis

The amount of DNA present in the cell was detected by Flow cytometry using propidium iodide (PI) staining. For each single test, $Pax7^{CreER}::Plk1^{f/f}$ myoblasts from three individual were harvest from a 10 cm dish after treated with 4-OHT or Methanol for 1 day. Then the myoblasts were washed in PBS for twice and fixed in pre-cooled 70% ethanol for overnight. After another two times of wash with PBS, myoblasts were centrifuged, and the pellet was resuspended with PBS containing 10 µg/ml ribonuclease. Suspended myoblasts were incubated at 37°C for 1 hr and the 10 µl (2 mg/ml) PI was added and incubated at 4°C for more than 1 hr before analysis using a BD flow cytometer.

Apoptosis of myoblasts was detected by Flow cytometry using Alexa Fluor 488 annexin V and PI (Invitrogen, V13241) double staining. For each single test, $Pax7^{CreER}::Plk1^{f/f}$ myoblasts from two individual were digested using trypsin from two 10 cm dishes after treated with 4-OHT or Methanol for 36 hr. Then the myoblasts were washed with pre-cooled PBS and centrifuged. The pellet was resuspended with 100 µl of 1X annexin-binding buffer, then 5 µL Alexa Fluor 488 annexin V and 1 µL 100 µg/mL PI working solution was added to each cell suspension. The mixtures were incubated at room temperature for 15 min then 400 µl 1X annexin-binding buffer was added before analysis using a BD flow cytometer.

## Isolation and culture of single fiber

Single fibers were isolated from extensor digitorum longus (EDL) muscles of adult mice as previously described (*Pasut et al., 2013*). Briefly, intact EDL muscles were digested in 0.2% type I collagenase (Sigma) in DMEM and incubated for approximately 1 hr at 37°C. Fibers were then liberated from the muscle bulk using graded glass pipettes. Suspended fibers were cultured in 60 mm horse-serum-coated plates in DMEM supplemented with 10% FBS, 4 ng/ml basic fibroblast growth factor (Promega), and 1% penicillin-streptomycin for 3 days. Freshly isolated fibers and cultured fibers were then fixed in 4% paraformaldehyde (PFA) for subsequent immunofluorescent analysis. MuSCs from at least 20 fibers were stained and count for statistical analysis.

## Hematoxylin–eosin and immunofluorescence staining

Fresh TA muscles and hind limb were embedded in optimal cutting temperature compound (OCT, Tissue-Tek) and frozen in isopentane that was chilled on dry ice. Frozen muscles were then cut into 10 µm-thick cross-sections by using a Leica CM1850 cryostat (Leica). For hematoxylin and eosin staining, the slides were first stained in hematoxylin for 15 min, rinsed in running tap water and then stained in eosin for 1 min. Slides were dehydrated in graded ethanol and Xylene, and then covered using Permount. Stained images were captured with a Nikon D90 digital camera installed on a Leica DM6000 (Leica) inverted microscope.

Immunofluorescence was performed on cross-sections, myofiber explants, primary myoblasts and differentiated myotubes. Briefly, samples were fixed in 4% PFA (paraformaldehyde) for 5 min and then permeabilized and blocked in PBS containing 5% goat serum, 2% bovine serum albumin (BSA), 0.2% Triton X-100, and 0.1% sodium azide for 1 hr. Samples were subsequently incubated with primary antibodies (Key resources table) overnight at 4°C. After washing with PBS, the samples were incubated with secondary antibodies and DAPI for 1 hr at room temperature. Fluorescent images were captured with a CoolSnap HQ charge coupled-device camera (Photometrics) by using a Leica

DM6000 microscope (Leica). 10 separated images were taken and count in each experimental groups.

## In situ TUNEL assay to detect cell apoptosis

For TUNEL and Pax7 staining, slides were fixed in 4% PFA for 10 min and then subjected to the TUNEL reaction using the CF488A TUNEL Assay Apoptosis Detection Kit (Biotium) according to the manufacturer's instructions. For negative control, samples were added TUNEL reaction buffer without TdT Enzyme. Samples treated with $H_2O_2$ (100 mM) for 30 min before TUNEL staining was set up as positive control. Counterstaining of Pax7 was then performed as regular immunofluorescence staining procedure.

## Quantitative real-time PCR

Total RNA from muscle tissue and myoblast was extracted by using Trizol reagent (Thermo Fisher Scientific). The first-strand cDNA was generated with random primer with MMLV Reverse Transcriptase (Thermo Fisher Scientific). Real-time PCR reactions were performed with a SYBR green PCR kit (Roche) in the Roche LightCycler 480 System (Roche). Primers for the genes of interest were all derived from the primer bank (Harvard Medical School) and were listed in *Supplementary file 2*. Gene expression was determined with the $2^{-\Delta\Delta Ct}$ relative quantification method and normalized to 18 s expression.

## Immunoblot analysis

Protein was extracted from homogenized muscle tissue or muscle cells (For Plk1-null myoblasts, each sample represent proteins extract from one 10 cm culture dish) with RIPA buffer that contained a protease inhibitor cocktail (Sigma) and phosphatase inhibitors NaF and $Na_3VO_4$. Protein concentration was measured using the BCA protein quantification kit (Pierce). Equal amounts of each protein sample were loaded for electrophoresis (Bio-Rad). Proteins were then transferred to a PVDF membrane (Biorad) and incubated with primary antibodies, followed by anti-rabbit or anti-mouse immunoglobulin G-horseradish peroxidase secondary antibody (Cell Signaling Technology). Signals were detected using fluorescence or chemiluminescence Western blot detection reagent (Santa Cruz Biotechnology) on a FluorChem E system (Protein Simple). Antibodies used for western blot analysis were listed in Key resources table.

## Single-cell gel electrophoresis

Single cell gel electrophoresis assay was performed in primary myoblast as previously described (*Collins, 2004*). Briefly, 10,000 cells were collected in 10 µl PBS and mixed with 75 µl of 0.5% low-melting point agarose. The cell-agarose mixture was then placed on a chilled and fully frosted slide with a 1% normal-melting point agarose coating layer. Subsequently, the slide was submerged in lysis solution (2.5 M NaCl, 100 mM $Na_2EDTA$, 10 mMTris, pH 10, 1% Sodium Sarcosinate with 1% triton X-100% and 10% DMSO being 1 hr before use) overnight at 4°C. Electrophoreses were carried out in alkaline electrophoresis buffer (1 mM $Na_2EDTA$ and 300 mM NaOH, pH >13) at 24 v, 300 milliamperes for 30 min. The slide was then neutralized and stained with 1 µg/ml DAPI for 15 min. Images were captured with CoolSnap HQ charge coupled-device camera (Photometrics) by using a Leica DM6000 microscope (Leica). 100 nuclei were measured by their tail DNA content for each treatment, and scored five classes (0–4) according to previous protocol (*Collins, 2004*). Class 0 represent 0% DNA is in tail and class 4 means 100% DNA is in the tail, respectively. Classes 1–3 are in between 0 and 100% and with an increment from 1 to 3.

## Statistical analysis

All analyses were conducted with Student's t-test (two-tail). All experimental data were presented as mean ± SEM. Comparisons with p values < 0.05 or <0.01 were considered statistically significant.

## Acknowledgements

This work was supported by grants from the US National Institutes of Health (R01AR071649) and the National Institute of Food and Agriculture (NC-1184). We thank Jun Wu and Mary Larimore for mouse colony maintenance, Xinyuan Xu, X Shawn Liu and Zhiguo Li for technical assistance.

## Additional information

### Funding

| Funder | Grant reference number | Author |
|---|---|---|
| National Institutes of Health | R01AR071649 | Feng Yue<br>Shihuan Kuang |
| U.S. Department of Agriculture | NC1184 | Shihuan Kuang |

The funders had no role in study design, data collection and interpretation, or the decision to submit the work for publication.

### Author contributions

Zhihao Jia, Data curation, Formal analysis, Validation, Investigation, Methodology, Writing—original draft, Writing—review and editing; Yaohui Nie, Data curation, Formal analysis, Investigation, Methodology, Writing—original draft, Writing—review and editing; Feng Yue, Yifan Kong, Lijie Gu, Investigation, Methodology, Writing—review and editing; Timothy P Gavin, Resources, Supervision, Writing—review and editing; Xiaoqi Liu, Conceptualization, Resources, Supervision, Writing—review and editing; Shihuan Kuang, Conceptualization, Resources, Formal analysis, Supervision, Funding acquisition, Writing—original draft, Project administration, Writing—review and editing

### Author ORCIDs

Zhihao Jia https://orcid.org/0000-0001-5525-2721
Shihuan Kuang https://orcid.org/0000-0001-9180-3180

### Ethics

Animal experimentation: Mouse maintenance and experimental use were performed according to Protocol #1112000440 approved by the Purdue University Animal Care and Use Committee.

### Decision letter and Author response

Decision letter https://doi.org/10.7554/eLife.47097.022
Author response https://doi.org/10.7554/eLife.47097.023

## Additional files

### Supplementary files

• Supplementary file 1. Genotypes and distribution of Plk1 conditional knockout embryos.
DOI: https://doi.org/10.7554/eLife.47097.018
• Supplementary file 2. Primers used in this study.
DOI: https://doi.org/10.7554/eLife.47097.019
• Transparent reporting form
DOI: https://doi.org/10.7554/eLife.47097.020

### Data availability

All data generated or analysed during this study are included in the manuscript and supporting files.

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
