## [Decision Letter]

Thank you for sending your article entitled "A requirement of Polo-like kinase 1 in murine embryonic myogenesis and adult muscle regeneration" for peer review at *eLife*. Your article is being evaluated by three peer reviewers, one of whom is a member of our Board of Reviewing Editors, and the evaluation is being overseen by a Reviewing Editor and Didier Stainier as the Senior Editor.

Given the list of essential revisions, including new experiments, the editors and reviewers invite you to respond within the next two weeks with an action plan and timetable for the completion of the additional work. We plan to share your responses with the reviewers and then issue a binding recommendation.

We are in the process of discussing the reviews. Before advising further and reaching a final decision, we would like to hear your response to the following concerns as provided by each reviewer, along with an estimated time frame for completing any additional work. Each reviewer has raised concerns regarding improved image quality and quantification for the same set of figures, along with additional experimentation to substantiate the conclusions as written. Of note, determining the fate of the Plk null cells and the cell cycle work described in Figures 5 and 6 was raised by all reviewers as being insufficient and will require additional analysis (as outlined) to buttress the conclusions.

*Reviewer #1:*

The manuscript of Jia et al. examines the role of polo-like kinase 1 (Plk1) in skeletal muscle stem cells (MuSCs). Notable strides have been made over the preceding decade in uncovering mechanisms that regulate MuSC commitment and differentiation, yet the regulatory controls that manage MuSC proliferation and survival remain largely unknown. Here, Jia et al. report that MuSC display a robust and transient spike in Plk1 expression, coincident to MuSC activation and proliferation (Figures 1 and 2). Subsequently, the authors conduct a series of intervention experiments to ascribe biologic function to Plk1 (Figures 3-7). Most of these experiments provide compelling outcomes, demonstrating that Plk1 is essential for the survival of both embryonic skeletal myoblasts and adult MuSCs (Figure 5-6), and that this survival mechanism operates (in part) through p53 upregulation (Figure 7). The data linking Plk1 to cell cycle progression is suggestive and fits with prior observations in the cancer field, yet this data is not as compelling as the pro-survival role characterized by the authors (see below for further discussion on this point). Taken together, this is an important set of observations that would be of considerable interest to skeletal muscle biologists and those that study cell survival/cell death in stem cell populations. However, at this stage there are a number of concerns the authors need to address to strengthen the conclusions as stated.

1) A reasonable conclusion that could be drawn from this data is that the primary function of Plk1 is to restrain caspase activation that occurs during MuSC differentiation (others have shown that caspase 3 activation is required for MuSc differentiation). This is entirely reasonable based on the data (and that fact that DNA damage has also been reported to be a downstream requirement of caspase induced differentiation, Larsen et al., 2010). As such, loss of Plk1 may 'release the brakes' on caspase 3, resulting in cell death rather than cell differentiation. Other kinases such as CK2 have been shown to act in this manner. The authors should include expanded discussion of this point in the manuscript.

2) As noted above the role of Plk1 in MuSC cell cycle progression is not compelling. In fact, the alteration in cell numbers may simply be the result of a change in cell survival (there are more cells because of the survival effect rather than true cell cycle progression). One way to address this issue would be to conduct a double thymidine block/release experiment on wildtype and Plk1 null cells and then measure thymidine incorporation, H2AX staining and comet assays over time. This would at least clarify the degree to which Plk1 influences survival vs. proliferation.

3) Given the concerns that caspase activity and DNA damage can mark differentiating cells as well as apoptotic cells, the authors need to include a more definitive cell death marker that is not open to interpretation. Annexin V/PI dual staining is a conclusive cell death measure and should be performed in a number of the experimental settings (primarily Figure 6). On this point, including FACS mediated PI staining as a measure of DNA content is misleading as it may be documenting dead cells rather than increased numbers of surviving cells (Figure 5F).

*Reviewer #2:*

A study examining the role of Plk1 in skeletal muscle stem cells (MuSCs) uses a combination of conditional ko mice, cultured cells and Plk1 inhibitors to attempt to establish that Plk1 is essential for myoblast proliferation and thus, required for skeletal muscle development and skeletal muscle regeneration. From further mechanistic studies the authors conclude that loss of Plk1 induced DNA damage and p53 up-regulation.

The data provided partially support the authors' conclusions. To fully support the conclusions made, additional experiments are needed. A further concern is that figures are incompletely labeled, figure legends are missing descriptions and required data are not provided including mouse sex, ages of mice or injury details. Throughout the manuscript there are mistakes, references to SCs that I assume means satellite cells and MuSCs that I assume are skeletal muscle stem cells. The authors should be consistent throughout to avoid confusion for the non-expert readers and carefully read and assess the text and legends for agreement, wording and typographical errors that are too numerous to address individually in the review.

For Figure 2 the authors claim that higher signal intensity of Plk1 colocalized to MyoD (subsection “Plk1 is dynamically expressed during muscle regeneration and myogenesis”, last paragraph). The image quality provided is insufficient to make this conclusion and no quantitative analyses were performed to substantiate this claim. Even though mRNA levels are higher in activated MuSCs the relationship of mRNA levels to protein abundance is complex and indirect.

I fail to understand how the authors conclude that Plk1 is necessary for Pax7 cell survival as Pax7 precedes MyoD (subsection “Loss of Plk1 in myogenic progenitors leads to embryonic lethality”, last paragraph). In Plk^MKO^ mice Plk1 is deleted upon MyoD expression and thus, determining a relationship between Plk1 and Pax7 is simply not possible. Similarly, in the aforementioned paragraph the claim that loss of Plk1 blocks proliferation and survival lacks the accompanying data to support the claim. The absence of MuSCs does not prove that Plk1 is necessary for survival. Furthermore, there appears to be muscle tissue in Figures 3C, 4B and 4D but the selected images are not accompanied by quantitative analyses that could include PCR or Western blots to confirm that no skeletal muscle is present and better support the authors' conclusions.

The data in Figure 4E are concerning as Pax7 immunoreactivity (red) is present in the interstitial areas, inside of myofibers, contained within nuclei and excluded from nuclei. Pax7 immunoreactivity is exclusively nuclear and while Pax7^+^ nuclei can be found in the interstitium and within myofibers during regeneration, there appears to be more Pax7 immunoreactivity outside of nuclei than contained within nuclei. In addition to Dystrophin immunoreactivity, the authors need to visualize Laminin to appropriately identify the MuSCs. The concerns noted for the immunofluorescence images question the validity of the quantitative data provided.

In the second paragraph of the subsection “Deletion of Plk1 impairs muscle regeneration in adult mice”, the authors make the claim that the absence of Pax7^+^ cells in the Plk1^PKO^ mice is due to proliferative failure. The cells are absent and without additional data I find it difficult to determine how the authors made this conclusion from the mere absence of cells. The inhibitor experiments that follow using BI2536 add little to the conclusions as the inhibitor will affect all cells that contain Plk1, which may comprise many different cell types in muscle tissue. In the last paragraph of the aforementioned subsection, the authors again state that Plk1 regulates proliferation with no supporting data but refer to Supplementary figure 3, whose legend does not describe the panels provided. The lack of adequate figure descriptions is particularly frustrating as the colors in the figure used for distinct immunoreactivity are not provided either in the figure itself or in the legend. I am not able to adequately interpret the figure.

Cells are stated to be synchronized for Figure 5 by a thymidine block, however, data to prove or support that the cells are indeed synchronized are not provided. How long do the cultures remain synchronized? What is the extent of synchronization? In Figure 5B is difficult to determine that the cells are on intact myofibers, the image quality is poor and in the lowest panel white is not defined and cannot be readily interpreted. For Figure 5F a complete cell cycle analysis is needed and not simply cells that are in G2/M. Has the timing of the cell cycle changed in the conditional ko cells? Again, I fail to understand how the data provided in Figure 5G confirm that the PKO cells undergo cell cycle arrest at mitosis. How have the authors determined whether or not cells are cycling from the data in this panel? I assume errors in the figure legend means arrows in the figure. Does methanol "induction" cause cell cycle arrest as well?

For Figure 6A the authors claim that 4OHT treatment causes all cells to apoptose based on Cas3 immunoreactivity. As there are only one or two cells that are Cas3^+^ in the figure how can the authors make the conclusions stated in the subsection “Plk1-null MuSCs apoptose”, for Figure 6B? Although the TUNEL labeling aids to support the conclusions regarding apoptosis, the authors should not publish Caspase 3 immunoreactivity as they clearly acknowledge that Caspase 3 is required for differentiation. Thus, despite the arguments made and the apparent lack of Myogenin immunoreactivity (Figure 6—figure supplement 1, lacking quantification) the rationale for using Caspase 3 is insufficient. in the aforementioned subsection, the authors state that all cells are Caspase 3 positive or TUNEL positive (Figure 6C-F) yet if one examines the panels provided there is a great deal of variability raising concerns that 100% of all cells are positive.

Without exhaustive details, overall, the quality of data in Figure 7 are insufficient to support the authors' conclusions, particularly those in panels A, B, and G. The inhibitor experiments are an initial method to support the conclusions, but the role of p53 would be best supported by a genetic experiment as a p53 heterozygous mouse should partially rescue the Plk1^PKO^ phenotype during muscle regeneration.

In summary, the rather low quality of many images, the lack of rigor in the text (the Supplementary figure 3 legend is wrong) missing labels in figures, and the lack of quantification in the data fail to support the authors' conclusions and provide a high level of frustration for me attempting to interpret data without adequate descriptions and labeling. Additional experimental details are lacking including the sex of mice, the precise ages as in 8 week old mice MuSCs are still fusing into myofibers, while in 12 wk old mice the fusion is at levels that have plateaued. In the injury experiments it is often unclear how long after CTX injection the muscle was harvested.

*Reviewer #3:*

The manuscript reports the function of Plk1 in muscle satellite cells both in developing and regenerating muscle. The study has novelty and a good sense of the possible mechanism of action of Plk1 in muscle stem cells. Some points could be better clarified to strengthen the data as suggested below.

In Figure 1, it would be more informative to study how the expression of PLK1 changes between proliferation and differentiation of myogenic cells. If indeed PLK1 is important for cell division, its expression levels should be increased in proliferative cultures compared to differentiated ones.

Additionally, the expression levels of different PLKs are assayed from regenerating tissue, which contains many cell types in addition to activated stem cells, such as immune cells, vascular progenitors and FAPs. It is unclear what the contribution of each cell type is to the overall levels of PLK1 observed.

In Figure 4—figure supplements 1 and 3 were the myofibers the same size in control versus PLK1^PKO^ mice?

In Figures 3 and 4, it appears that very few muscle fibers are formed in the PLK1-deficient muscle, yet many mononuclear cells are present. It seems important to define the identity of these cells, whether they are progenitors of satellite cells or if they are non-myogenic cells replacing the space of myogenic cells.

In Figure 4 the data shown following cardiotoxin injury is 7d post injection. Do muscles look the same at 21 days following injury as they do at 7 days? In other words, are muscle differentiation/repair delayed due to loss of PLK1 or inhibited/arrested as they are at 7 days?

In Figure 5F the P2 gate includes cells in S phase and there is an additional peak/shoulder indicating the cells in G2/M that is more prominent in the PLK1^PKO^. The exact conditions used should be described as 'doublets' (which are normally excluded by FACS operators) should be included and will provide meaningful data.

One concern with the analyses presented in Figures 6-7 is the number of cells that were obtained or used to perform these analyses. From the data provided in previous figures, the conclusion is that cells are arrested in G2/M and do not divide. If this is the case and cell division is the main cellular defect observed, how were sufficient cell numbers obtained to perform analyses such as western blot? In addition, if many of the cells undergo apoptosis, how robust were these analyses in terms of total cell numbers?

---

## [Author Response]

[Editors' note: the authors’ plan for revisions was approved and the authors made a formal revised submission.]

Reviewer #1:[…] 1) A reasonable conclusion that could be drawn from this data is that the primary function of Plk1 is to restrain caspase activation that occurs during MuSC differentiation (others have shown that caspase 3 activation is required for MuSc differentiation). This is entirely reasonable based on the data (and that fact that DNA damage has also been reported to be a downstream requirement of caspase induced differentiation, Larsen et al., 2010). As such, loss of Plk1 may 'release the brakes' on caspase 3, resulting in cell death rather than cell differentiation. Other kinases such as CK2 have been shown to act in this manner. The authors should include expanded discussion of this point in the manuscript.

That’s a great point. We have now provided new data to support the potential role of Plk1 inhibition induced Caspase-3 activation in promoting myogenic differentiation. While deletion of Plk1 during proliferation stage leads to cell death and precludes the opportunity to examine its effect in differentiation, we found that inhibition of Plk1 kinase activity using BI2536 promotes myogenic differentiation. We included these data in Figure 6—figure supplement 2 and also added discussion in the fifth paragraph of the Discussion section.

2) As noted above the role of Plk1 in MuSC cell cycle progression is not compelling. In fact, the alteration in cell numbers may simply be the result of a change in cell survival (there are more cells because of the survival effect rather than true cell cycle progression). One way to address this issue would be to conduct a double thymidine block/release experiment on wildtype and Plk1 null cells and then measure thymidine incorporation, H2AX staining and comet assays over time. This would at least clarify the degree to which Plk1 influences survival vs. proliferation.

From the single fiber culture in Figure 5B and Figure 6A, B, all the Plk1-null MuSCs lost the ability to proliferate, stopped before completing the first division and die within 72h (also quantified in Figure 5—figure supplement 1). As suggested, we have now performed the double thymidine block/release experiment on wildtype and Plk1 null cells (Figure 7—figure supplement 2A-B). Our new results support that Plk1-null myoblasts were arrested at M-phase and undergo DNA damage response induced apoptosis.

3) Given the concerns that caspase activity and DNA damage can mark differentiating cells as well as apoptotic cells, the authors need to include a more definitive cell death marker that is not open to interpretation. Annexin V/PI dual staining is a conclusive cell death measure and should be performed in a number of the experimental settings (primarily Figure 6). On this point, including FACS mediated PI staining as a measure of DNA content is misleading as it may be documenting dead cells rather than increased numbers of surviving cells (Figure 5F).

As suggested, we have now substantiated the conclusion by Annexin V-FITC Apoptosis Staining, and the results were shown in Figure 6—figure supplement 1C-D. Our new results shown that Plk1-null myoblasts indeed are mostly labeled by the Annexin V, while the control cells were mostly negative. We agree that PI staining is a common method to distinguish live and dead cells as live cells exclude PI and thus make them unlabeled, while dead cells are positively labeled. However, our study used PI on fixed dead cells, where all cells are labeled and the labeling intensity correlates to DNA content. This is a commonly used method for FACS based determination DNA content in fixed cells. We also performed TUNEL Assay as a readout of apoptosis.

Reviewer #2:A study examining the role of Plk1 in skeletal muscle stem cells (MuSCs) uses a combination of conditional ko mice, cultured cells and Plk1 inhibitors to attempt to establish that Plk1 is essential for myoblast proliferation and thus, required for skeletal muscle development and skeletal muscle regeneration. From further mechanistic studies the authors conclude that loss of Plk1 induced DNA damage and p53 up-regulation.The data provided partially support the authors' conclusions. To fully support the conclusions made, additional experiments are needed. A further concern is that figures are incompletely labeled, figure legends are missing descriptions and required data are not provided including mouse sex, ages of mice or injury details. Throughout the manuscript there are mistakes, references to SCs that I assume means satellite cells and MuSCs that I assume are skeletal muscle stem cells. The authors should be consistent throughout to avoid confusion for the non-expert readers and carefully read and assess the text and legends for agreement, wording and typographical errors that are too numerous to address individually in the review.

Thank you for the advice. As suggested, we have gone through the main text, figure labels, legends, and descriptions, and fixed typographical errors. We have also provided detailed information on animals used (subsection “Mice and animal care”). In addition, the detailed information of animals used in every experiment was also added in the revised figure legend. MuSCs (for Muscle Satellite Cells) were used throughout the revised manuscript.

For Figure 2 the authors claim that higher signal intensity of Plk1 colocalized to MyoD (subsection “Plk1 is dynamically expressed during muscle regeneration and myogenesis”, last paragraph). The image quality provided is insufficient to make this conclusion and no quantitative analyses were performed to substantiate this claim. Even though mRNA levels are higher in activated MuSCs the relationship of mRNA levels to protein abundance is complex and indirect.

Thank you for pointing this out. Our original statement was not accurate, and we meant to say that Plk1 signal was co-localized to MyoD signal. We replaced the images with a higher magnification one and rephrased the statement. We also counted over 200 cells on myofibers cultured for 72h, all Plk1^+^ cells were also MyoD^+^. The results were updated in the last paragraph of the subsection “Plk1 is dynamically expressed during muscle regeneration and myogenesis”.

I fail to understand how the authors conclude that Plk1 is necessary for Pax7 cell survival as Pax7 precedes MyoD (subsection “Loss of Plk1 in myogenic progenitors leads to embryonic lethality”, last paragraph). In Plk^MKO^ mice Plk1 is deleted upon MyoD expression and thus, determining a relationship between Plk1 and Pax7 is simply not possible. Similarly, in the aforementioned paragraph the claim that loss of Plk1 blocks proliferation and survival lacks the accompanying data to support the claim. The absence of MuSCs does not prove that Plk1 is necessary for survival. Furthermore, there appears to be muscle tissue in Figures 3C, 4B and 4D but the selected images are not accompanied by quantitative analyses that could include PCR or Western blots to confirm that no skeletal muscle is present and better support the authors' conclusions.

The statement “Plk1 is necessary for Pax7 cell survival” was based on the understanding that Pax7^+^ progenitors should have been generated before turning on Myod expression, which then drives deletion of Plk1. Reading this statement again, without the results that were shown later in the manuscript, we agree that the sentence is misleading. We also agree that there were no data to directly support “proliferation and survival” here. We have therefore reworded this phrase as “Plk1 is necessary for the generation or maintenance of embryonic myoblasts”.

To answer the second part regarding whether there is muscle tissue in Figures 3C, 4B and 4D, we must first clarify that these figures are from two different mice models. In the Plk1^MKO^ mice, Plk1 is deleted in MyoD-expressing embryonic myoblasts. Figure 3C is based on this model and we do not see any eMyHC marked myotubes (shown in Figure 3D, from the same limbs shown in Figure 3C). The second model is Plk1^PKO^ mice, in which Plk1 is deleted in Pax7-expressing cells when tamoxifen or 4-OH-Tamoxifen is administered. Figure 4B and Figure 4D are from the latter mouse model in which muscles were allowed to develop normally and then tamoxifen was administered to delete Plk1 only in the Pax7^+^ satellite cells. Again, we don’t see any central nucleated dystrophin+ myofibers (Figure 4E) that were regenerated after degeneration of the original muscles prior to deletion of Plk1.

The data in Figure 4E are concerning as Pax7 immunoreactivity (red) is present in the interstitial areas, inside of myofibers, contained within nuclei and excluded from nuclei. Pax7 immunoreactivity is exclusively nuclear and while Pax7^+^ nuclei can be found in the interstitium and within myofibers during regeneration, there appears to be more Pax7 immunoreactivity outside of nuclei than contained within nuclei. In addition to Dystrophin immunoreactivity, the authors need to visualize Laminin to appropriately identify the MuSCs. The concerns noted for the immunofluorescence images question the validity of the quantitative data provided.

It is not uncommon to see non-specific (non-nuclear) signals in Pax7 staining as the antibody was murine origin and the secondary antibody used would recognize endogenous mouse IgG1. To ensure scientific rigor, our quantification only include the nuclear signals (i.e. DAPI and Pax7 double positive signals), and all non-nuclear signals were excluded in the quantification. We have now included the description in the revised manuscript (subsection “Deletion of Plk1 impairs muscle regeneration in adult mice”, first paragraph).

In the second paragraph of the subsection “Deletion of Plk1 impairs muscle regeneration in adult mice”, the authors make the claim that the absence of Pax7^+^ cells in the Plk1^PKO^ mice is due to proliferative failure. The cells are absent and without additional data I find it difficult to determine how the authors made this conclusion from the mere absence of cells. The inhibitor experiments that follow using BI2536 add little to the conclusions as the inhibitor will affect all cells that contain Plk1, which may comprise many different cell types in muscle tissue. In the last paragraph of the aforementioned subsection, the authors again state that Plk1 regulates proliferation with no supporting data but refer to Supplementary figure 3, whose legend does not describe the panels provided. The lack of adequate figure descriptions is particularly frustrating as the colors in the figure used for distinct immunoreactivity are not provided either in the figure itself or in the legend. I am not able to adequately interpret the figure.

We agree and rephrased the sentence and deleted “proliferative failure”, though the inhibitor and KO experiments performed in cultured myoblasts later (Figure 5—figure supplement 1 and Figure 5—figure supplement 2) showed cell autonomous effects. We apologize for the confusing mis-matched figure legends (for several supplementary figures) due to rearrangement of the order of figures before the submission (the Supplementary figure 1 legend should be corresponding to Figure 4—figure supplement 3, Supplementary figure 2 legends correspond to Figure 4—figure supplement 1; and Supplementary figure 3 legends correspond to Figure 4—figure supplement 2). These are corrected in the revised manuscript. Colors used for immunoreactivity were also labeled in the revised Figure 4—figure supplement 3 legend.

Cells are stated to be synchronized for Figure 5 by a thymidine block, however, data to prove or support that the cells are indeed synchronized are not provided. How long do the cultures remain synchronized? What is the extent of synchronization? In Figure 5B is difficult to determine that the cells are on intact myofibers, the image quality is poor and in the lowest panel white is not defined and cannot be readily interpreted. For Figure 5F a complete cell cycle analysis is needed and not simply cells that are in G2/M. Has the timing of the cell cycle changed in the conditional ko cells? Again, I fail to understand how the data provided in Figure 5G confirm that the PKO cells undergo cell cycle arrest at mitosis. How have the authors determined whether or not cells are cycling from the data in this panel? I assume errors in the figure legend means arrows in the figure. Does methanol "induction" cause cell cycle arrest as well?

We used the mitosis marker phosphohistone H3 (pHH3) to show the synchronization efficiency, where the cell peaked at M-Phase 14 hours after the release (Figure 5A). Cells are typically synchronous in the first cycle after release and we did not follow cells after 20 hours as it is beyond the first cycle in the WT cells, when they become asynchronous. The host myofbers were not seen as they are out of focus. The white channel in Figure 5B represents the DAPI staining and we added the label in revised manuscript. We also performed analyses at more time points after double thymidine block to pinpoint defects prior to G2/M, but stages after G2/M is impossible as the KO cells don’t seem to pass that time point. In Figure 5F, we used PI staining to show cells containing doubled DNA content suggestive of completion of S-phase. In Figure 5G, the white arrows indicate cells containing unsegregated chromosomes, which were more abundant in KO cells than control (methanol treated) cells. We have updated the figure legend and we don’t think 0.1% of methanol caused cell cycle arrest.

For Figure 6A the authors claim that 4OHT treatment causes all cells to apoptose based on Cas3 immunoreactivity. As there are only one or two cells that are Cas3^+^ in the figure how can the authors make the conclusions stated in the subsection “Plk1-null MuSCs apoptose”, for Figure 6B? Although the TUNEL labeling aids to support the conclusions regarding apoptosis, the authors should not publish Caspase 3 immunoreactivity as they clearly acknowledge that Caspase 3 is required for differentiation. Thus, despite the arguments made and the apparent lack of Myogenin immunoreactivity (Figure 6—figure supplement 1, lacking quantification) the rationale for using Caspase 3 is insufficient. in the aforementioned subsection, the authors state that all cells are Caspase 3 positive or TUNEL positive (Figure 6C-F) yet if one examines the panels provided there is a great deal of variability raising concerns that 100% of all cells are positive.

There were few myoblasts on cultured myofibers because the MuSCs could not complete the first division and then undergo apoptosis. Nevertheless, the statement that all myoblasts were Casp3^+^ or TUNEL^+^ was based on observation of at least 30 cells from 30 myofibers from 3 mice. Please note that the MyoD^–^ cells on myofibers are differentiated myonuclei that were not cycling and excluded from the analysis.

For Figure 6C-F, after 48h of 4-OHT treatment, most Plk1-null myoblasts were dead and floated, and the remainder cells were either Cleaved-caspase3^+^ or TUNEL^+^ based our staining. Myoblast treated with 100 mm H_2_O_2_ for 30mins were stained as a positive control signal. All images were captured and handled using the same parameters as used in the positive control. Statistical analysis for quantification was based on 25-50 myoblasts from 3 replicates from 3 individual mice. In addition, the new double thymidine treatment data (Figure 7—figure supplement 2A-B) also indicated that after 16-24 h double thymidine block/release, all 4-OHT treated myoblasts were TUNEL^+^.

Without exhaustive details, overall, the quality of data in Figure 7 are insufficient to support the authors' conclusions, particularly those in panels A, B, and G. The inhibitor experiments are an initial method to support the conclusions, but the role of p53 would be best supported by a genetic experiment as a p53 heterozygous mouse should partially rescue the Plk1^PKO^ phenotype during muscle regeneration.

We appreciate the comment but establishing a conditional KO on the p53 null or heterozygous background requires breeding of three transgenic alleles into one mouse that will require extended time (at least 6 months) and is beyond the scope of this manuscript. Nevertheless, we have added discussion on the reported role of p53 on MuSCs survival and cell cycle progression to address this comment (Discussion, fifth paragraph).

In summary, the rather low quality of many images, the lack of rigor in the text (the Supplementary figure 3 legend is wrong) missing labels in figures, and the lack of quantification in the data fail to support the authors' conclusions and provide a high level of frustration for me attempting to interpret data without adequate descriptions and labeling. Additional experimental details are lacking including the sex of mice, the precise ages as in 8 week old mice MuSCs are still fusing into myofibers, while in 12 wk old mice the fusion is at levels that have plateaued. In the injury experiments it is often unclear how long after CTX injection the muscle was harvested.

We are sorry for these confusions and hope our revised manuscript addressed these concerns.

Reviewer #3:The manuscript reports the function of Plk1 in muscle satellite cells both in developing and regenerating muscle. The study has novelty and a good sense of the possible mechanism of action of Plk1 in muscle stem cells. Some points could be better clarified to strengthen the data as suggested below.In Figure 1, it would be more informative to study how the expression of PLK1 changes between proliferation and differentiation of myogenic cells. If indeed PLK1 is important for cell division, its expression levels should be increased in proliferative cultures compared to differentiated ones.

Thank you for the suggestion. The comparison of Plk1 level in proliferation and differentiation myoblasts were shown in Figure 1D, where cells were in proliferation stage at Day 0. We clarified this in the revised manuscript (subsection “Plk1 is dynamically expressed during muscle regeneration and myogenesis”, first paragraph).

Additionally, the expression levels of different PLKs are assayed from regenerating tissue, which contains many cell types in addition to activated stem cells, such as immune cells, vascular progenitors and FAPs. It is unclear what the contribution of each cell type is to the overall levels of PLK1 observed.

We added a sentence to acknowledge the potential contribution of non-MuSCs to the overall changes in Plk1 levels in the whole muscle (subsection “Plk1 is dynamically expressed during muscle regeneration and myogenesis”, last paragraph). However, the data in Figure 2B comparing freshly FACS-sorted MuSCs that are not yet cycling to cells that were activated by culturing for 24 hours shows that the Plk1 level is increased by 15-fold upon activation. This observation strongly suggests that Plk1 level in MuSC contributed at least in part to the overall change in the muscle.

In Figure 4—figure supplements 1 and 3 were the myofibers the same size in control versus PLK1^PKO^ mice?

We have now quantified fiber size in the revised version (Figure 4—figure supplement 1D). For Figure 4—figure supplement 2, as there were no intact muscle fibers in CTX treated Plk1^PKO^ mice, it would be impossible to quantify the myofiber size.

In Figures 3 and 4, it appears that very few muscle fibers are formed in the PLK1-deficient muscle, yet many mononuclear cells are present. It seems important to define the identity of these cells, whether they are progenitors of satellite cells or if they are non-myogenic cells replacing the space of myogenic cells.

Based on the Pax7 staining in Figure 3 and 4, no MuSCs were found in Plk1-deficient muscle. We have now used markers of adipocytes (PLIN1) and macrophages (F4/80) to demonstrate that some of these cells are infiltrating macrophages and adipocytes (Figure 4—figure supplement 2I).

In Figure 4 the data shown following cardiotoxin injury is 7d post injection. Do muscles look the same at 21 days following injury as they do at 7 days? In other words, are muscle differentiation/repair delayed due to loss of PLK1 or inhibited/arrested as they are at 7 days?

Our data showing the complete lack of regenerated myofibers and Pax7^+^ cells indicated a complete regenerative failure (but not delay). As suggested, we have now performed analysis of muscle regeneration at 21 DPI and the results exclude the possibility of regenerative delay (Figure 4—figure supplement 2F-I).

In Figure 5F the P2 gate includes cells in S phase and there is an additional peak/shoulder indicating the cells in G2/M that is more prominent in the PLK1^PKO^. The exact conditions used should be described as 'doublets' (which are normally excluded by FACS operators) should be included and will provide meaningful data.

Thank you for pointing these out and we now refer these cells to tetraploid cells (they are single cells with DNA doubled) in the revised manuscript. (subsection “Plk1-null MuSCs undergo cell cycle arrest at M Phase”, first paragraph).

One concern with the analyses presented in Figures 6-7 is the number of cells that were obtained or used to perform these analyses. From the data provided in previous figures, the conclusion is that cells are arrested in G2/M and do not divide. If this is the case and cell division is the main cellular defect observed, how were sufficient cell numbers obtained to perform analyses such as western blot? In addition, if many of the cells undergo apoptosis, how robust were these analyses in terms of total cell numbers?

Figures 6-7 each contains two sets of experiments: one performed on single myofibers in culture and another set on primary myoblasts from Plk1^PKO^ mice without prior tamoxifen administration to delete Plk1. These cells were allowed to grow normally and then acutely deleted of Plk1 by 4-OH-Tamoxifen administration. Cell numbers used for analysis were listed as below:

1. For fibers, 30 single fibers from 3 mice (10 each) were used for quantification.

2. Immunofluorescence of Cleaved Caspase3, TUNEL and γH2AX were quantified based on 25-50 myoblasts (each) from 3 replicates from 3 mice.

3. In order to collect enough protein for WB analysis, myoblasts from a 10-cm culture dish were used in each sample (more than one million myoblasts). In addition, they were treated with 4-OHT for shorter time to make sure that most of them were not dead and floated. The information was updated in the revised manuscript (subsection “Immunoblot analysis”).